# Country adoption of WHO 2019 guidance on HIV testing strategies and algorithms: a policy review across the WHO African region

Emmanuel Fajardo [ORCID],[1] Céline Lastrucci,[1] Nayé Bah,[2]
Casimir Manzengo Mingiedi,[3] Ndoungou Salla Ba,[4] Fausta Mosha,[5]
Frank John Lule,[6] Margaret Alia Samson Paul,[6] Lago Hughes,[6]
Magdalena Barr-DiChiara,[1] Muhammad S Jamil,[1] Anita Sands,[7] Rachel Baggaley,[1]
Cheryl Johnson[1]

For numbered affiliations see end of article.

**Correspondence to**
Dr Cheryl Johnson;
johnsonc@who.int

## ABSTRACT

**Objectives** In 2019, the WHO released guidelines on HIV testing service (HTS). We aim to assess the adoption of six of these recommendations on HIV testing strategies among African countries.

**Design** Policy review.

**Setting** 47 countries within the WHO African region.

**Participants** National HTS policies from the WHO African region as of December 2021.

Primary and secondary outcome measures: Uptake of WHO recommendations across national HTS policies including the standard three-test strategy; discontinuation of a tiebreaker test to rule in HIV infection; discontinuation of western blotting (WB) for HIV diagnosis; retesting prior to antiretroviral treatment (ART) initiation and the use of dual HIV/syphilis rapid diagnostic tests (RDTs) in antenatal care. Country policy adoption was assessed on a continuum, based on varying levels of complete adoption.

**Results** National policies were reviewed for 96% (n=45/47) of countries in the WHO African region, 38% (n=18) were published before 2019 and 60% (n=28) adopted WHO guidance. Among countries that had not fully adopted WHO guidance, not yet adopting a three-test strategy was the most common reason for misalignment (45%, 21/47); of which 31% and 22% were in low-prevalence (<5%) and high-prevalence (≥5%) countries, respectively. Ten policies (21%) recommended the use of WB and 49% (n=23) recommended retesting before ART initiation. Dual HIV/syphilis RDTs were recommended in 45% (n=21/47) of policies.

**Conclusions** Many countries in the African region have adopted WHO-recommended HIV testing strategies; however, efforts are still needed to fully adopt WHO guidance. Countries should accelerate their efforts to adopt and implement a three-test strategy, retesting prior to ART initiation and the use of dual HIV/syphilis RDTs.

## INTRODUCTION

HIV testing service (HTS) is the critical gateway to accessing HIV-related care and treatment for those diagnosed as HIV-positive and as a means to accessing prevention services for those who test HIV-negative.[1 2] Despite considerable progress, reaching the Joint Unite Nation's Programme on HIV/AIDS (UNAIDS) 95-95-95 targets by 2025 will require increased testing uptake, particularly among populations with testing gaps such as key population, youth and men.[3] Further, the global emergency caused by the COVID-19 pandemic in 2020 has disrupted many health services including HTS, and supply chains of key commodities such as rapid diagnostic tests (RDTs) have been stretched.[4] In 2021, it was estimated that 85% of people living with HIV globally were aware of their serostatus.[5]

In 2019, the WHO published consolidated guidelines on HTS,[6] bringing together existing and new recommendations specific

---

### STRENGTHS AND LIMITATIONS OF THIS STUDY

⇒ This study provided a comprehensive review of existing national HIV testing service policies in the African region and was able to identify current policies for 45 countries.

⇒ Data were collected using a robust search and global repository developed by WHO, as well as a standardised extraction tool to synthesise national policy information.

⇒ The review focused on published policies and did not assess country implementation at the national, subnational or site level.

⇒ The review focused on WHO recommendations related to HIV testing strategies for individuals ≥18 months of age.

⇒ The review identified African countries which reported that they were in the process of adopting WHO recommendations—particularly the use of a three-test strategy and dual HIV/syphilis rapid diagnostic tests.

to testing strategies and algorithms. This includes existing guidance on the use of serial testing rather than parallel testing (WHO, 2012),[7] discontinuation of a tiebreaker strategy to rule-in HIV infection (WHO, 2015) [1] and retesting prior to antiretroviral treatment (ART) initiation, recommended since 2014.[1] As well as three new recommendations on: (1) the use of a standard three-test strategy which uses three consecutive HIV-reactive tests to provide and HIV-positive diagnosis, (2) the discontinuation of western blot (WB) and line immunoassays (LIA) and (3) the use of dual HIV/syphilis RDTs as first assay in HTS in antenatal care (ANC). In light of these testing modalities, ensuring that HIV testing is accessible and adheres to the '5 C's' including voluntary consent, confidentiality, counselling, correct results and linkage to care, is of utmost importance.

These recommendations have been prioritised because they are essential to achieving global 95-95-95 goals by making a testing accurate, affordable and high impact.[3 6] By moving away from testing services with WB, programmes will no longer have long turnaround times which delay the ability to offer same-day ART or pre-exposure prophylaxis (PrEP) initiation.[6] The adoption of innovative tools such as the dual HIV/syphilis RDTs will enable more people to be tested and treated for syphilis, which is essential for achieving triple elimination goals.[6] Additionally, many countries still use the 2015 WHO recommendation of using the national HIV prevalence to determine whether a two-assay (≥5%) or three-assay testing strategy (<5%) should be used.[1] However, as ART coverage has expanded HIV positivity and the proportion of people with HIV who are undiagnosed and not in care will continue to decline. As a result, the positive predictive value of previous testing strategies will also decline and lead to an increase in false positive diagnoses. In relation to the guideline development process, WHO now recommends all countries use a standard three-test strategy.[8] WHO also recommends that countries planning to update their HIV testing algorithms undertake a verification study to select appropriate HIV serology products and ensure they do not cross-react in order to minimise the risk of misdiagnosis.[9]

To support the implementation of the 2019 guidelines, WHO launched and disseminated the guidance at the Africa Society of Laboratory Medicine and the International Conference on AIDS and STIs in Africa. Following this, WHO also provided detailed country support to adopt the guidelines along with developing an application to access the guidelines more easily. In addition to dissemination and country support,[10] it is critical to monitor and track the implementation of WHO policy uptake to understand policy priorities and challenges overtime. Such tracking not only provides valuable insights but can help guide revisions to future guidance and can help target country policy support.[11 12]

A global review of national HIV testing policies was conducted by WHO in 2018 to assess the adoption of its testing recommendations and policies.[13] Of 91 policies reviewed, only 24 (25%) adhered to WHO guidance. This policy review follows on from the previous review and seeks to assess country's uptake of six 2019 WHO recommendations related to HIV testing strategies. Here, we focus on the WHO African region, which hosts several countries with a high HIV-burden and with large HIV testing programmes. The same policy review across the remaining five WHO regions is underway as part of the update to the 2023 WHO HTS guidelines.

## METHODS

### Search strategy

We carried out a comprehensive desk review of national HIV testing policies in 47 Member States in the WHO African region (online supplemental figure 1) using a policy repository maintained by WHO, scanning government websites and contacting government officials or other relevant informants through December 2021. The most current policy documents containing information on the HTS policy were included, but when this was unavailable, we included the most recent HIV testing algorithm, if that was also unavailable, we opted to include previous national policies. Documents of all languages were included. Other supporting documents related to algorithm validation and PMTCT guidelines were identified, when possible, either through references provided in the national HTS policy or through contact with key informants. The full protocol for the review was previously developed and published in 2018.[13]

### Data extraction

Data were extracted from each policy document by one reviewer (EF) into standardised coding forms on policy information, HIV testing strategy/algorithm (>18 months of age), dual HIV/syphilis testing strategy/algorithm and retesting prior to ART initiation (online supplemental table 1). When reported the order, type of test kits and name brand of diagnostics used within the HIV testing algorithm was summarised descriptively, as well as information on whether the algorithm was verified or validated.

A second reviewer (CL) carried out cross-checking of the data. Differences between coders were resolved through a third reviewer (CJ). To prevent misclassification, items were marked as 'unclear' during data extraction when a lack of information prevented complete understanding. Reviewers then worked to contact key informants to provide further detail and clarity wherever possible.

### Analysis

We assessed national adoption of WHO HTS guidance using six specific recommendations related to testing strategies and algorithms for those ≥18 months of age outlined in the WHO 2019 HIV testing guidelines, namely: (1) use of serial testing, (2) use of a three-test strategy, (3) discontinuation of a tiebreaker to rule in HIV infection, (4) discontinuation of WB/LIA, (5) retesting prior to ART initiation and (6) use of dual HIV/syphilis RDT

in ANC. Additional qualitative details about the national testing algorithm, such as the order of and name brand of test kits used in testing algorithms, were also reviewed when reported and assessed according to WHO recommendations (see box 1).

Box 1  **WHO recommendations for HIV testing strategies for individuals ≥18 months of age, 2019**

⇒ In response to changes in the HIV epidemic, WHO recommends countries use three consecutive reactive tests to provide an HIV-positive diagnosis.

⇒ Testing strategies should use tests serially, not in parallel and should not use a tiebreaker to rule-in HIV infection. Instead, those with discrepant test results should be ruled inconclusive and referred for further testing in 14 days to rule-in or rule-out seroconversion.

⇒ WHO recommends that all HIV testing algorithms use a combination of RDTs and/or enzyme-immunoassays (EIAs) to achieve at least 99% positive predictive value and use a combination of tests with ≥99% sensitivity and ≥98% specificity. WHO does not recommend the use of nucleic acid testing techniques within HIV testing algorithms for individuals ≥18 months of age.

⇒ The first test in an HIV testing strategy and algorithm should have the highest sensitivity, followed by a second and third test of the highest specificity. Algorithms should be validated and verified to ensure high-quality and accurate testing.

⇒ All people newly diagnosed with HIV should be retested to verify their HIV status prior to starting ART, using the same testing strategy and algorithm as the initial test. To minimise the risk of misdiagnosis, this approach should be maintained in settings in which rapid ART initiation is being implemented.

Source: WHO 2019.[6]

Based on the number of recommendations adopted, national policies were categorised as: adopted (6); nearly adopted (5–4); somewhat not adopted: (3) and not adopted: (2 or less). We then provide simplified reporting (adopted, partially adopted and not adopted) for national policies which reviewed in 2018 and 2021 to assess changes overtime. Notably, these categories were initially developed as part of the 2018 policy review and maintained to assist with future updates and policy tracking.

Descriptive analyses were then stratified by subregions (Western, Central, Eastern and Southern Africa; online supplemental figure 1) to determine rates of adherence. All analyses were conducted in Microsoft Excel. Countries that had policies reviewed in 2018 and 2021 were also compared with assess changes in alignment with WHO recommendations over time.

### Patient and public involvement
Patients and/or the public were not involved in the design, conduct, reporting or dissemination plans of this study.

### RESULTS
In total, we were able to identify policy documents from 96% of countries from the WHO African region (n=45/47); 71% of countries had a policy reviewed in both 2018 and 2021 (n=32/45). Two countries, Republic of the Congo and Cabo Verde, were not included because we were unable to obtain their national policies and sufficient information (figure 1). Out of all these policies, 91% (41/45) had updated their national policies as of December 2021. Policy publication dates ranged from 2013 to 2021. Out of the six country policies that had not been fully updated since 2018 (Algeria, Botswana, eSwatini, Cote d'Ivoire, Malawi and Zimbabwe), one country (Zimbabwe) provided a new HIV testing algorithm that was included in the review.

Of the 45 policies providing information on HIV testing strategies, 16 were from Western Africa (36%), 16 were from Eastern Africa (36%), 8 were from Central Africa (18%) and 5 were from Southern Africa (11%). Policies were published in English (n=22), French (n=18), Portuguese (n=4) and Spanish (n=1). Based on the most recent national HIV prevalence reported by UNAIDS,[1] 34 countries (76%) had a low HIV prevalence (<5%) and 11 countries (24%) had a high HIV prevalence (≥5%) (online supplemental table 2).

### Overall adoption of WHO-recommended HIV testing strategies
In 2021, 59% of national testing strategies (n=28/45) were either fully or partially aligned with WHO 2019 HTS recommendations and 36% (n=17/47) had not adopted the recommendations (table 1). When the analysis was restricted to a subset of countries with policies in

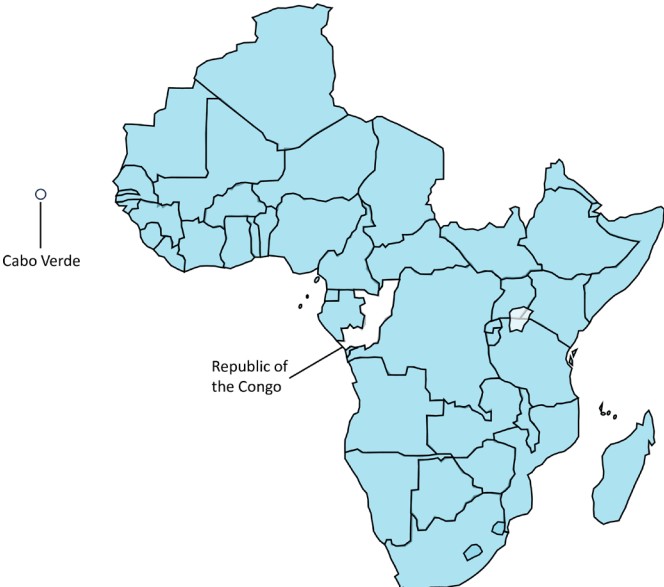

**Figure 1**  Countries from the WHO African Region with HIV testing policies identified (in blue) and included in the analysis (n=45/47). The two countries not included in the review are indicated in the map.

**Table 1** Policy adoption of 2019 WHO-recommended HIV testing strategies, WHO regional office for Africa (AFRO), 2021

| AFRO subregions (countries) | Policies reviewed | Serial testing strategy | Discontinuation of tiebreaker* testing strategy | Retesting prior to ART initiation | Recommended three-test strategy† | Use of dual HIV/syphilis test‡ | No use of Western Blot |
|---|---|---|---|---|---|---|---|
| All (n=47) | 45 (96%) | 29 (62%) | 37 (79%) | 23 (49%) | 21 (45%) | 21 (45%) | 35 (74%) |
| Western (n=17) | 16 (94%) | 9 (53%) | 9 (53%) | 5 (29%) | 8 (47%) | 7 (41%) | 10 (59%) |
| Central (n=9) | 8 (89%) | 7 (78%) | 7 (78%) | 5 (56%) | 3 (33%) | 2 (22%) | 6 (67%) |
| Eastern (n=16) | 16 (100%) | 12 (75%) | 16 (100%) | 9 (56%) | 9 (56%) | 9 (56%) | 14 (88%) |
| Southern (n=5) | 5 (100%) | 1 (20%) | 5 (100%) | 4 (80%) | 1 (20%) | 3 (60%) | 5 (100%) |

*Use of a third assay to rule-in HIV infection.
†At the time of the review, some countries noted that they were in the process of transitioning to a three-test strategy; however, they were not included as in compliance because their national guidance and testing strategy was not yet finalised.
‡Dual HIV/syphilis RDT as the first test in ANC. Note that at the time of the review, five countries were using test outside of WHO guidance. Two countries had non-compliant algorithms (Lesotho and Zambia) and three countries (Liberia, Madagascar and Uganda) were also using the test among key populations and partners of pregnant women. As of 2023, however, the dual test is recommended among key populations and those with high ongoing risk.
ANC, antenatal care; RDT, rapid diagnostic test.

both the 2018 and 2021 review (n=32), the proportion adopting WHO guidance in 2018 was 28% (n=9/32) vs 62% (n=20/32) in the current (2021) review (see online supplemental table 3, online supplemental figure 2). Across the subregions, Southern (80%) and Eastern (75%) Africa had the greatest level of alignment, while Central Africa (63%) and Western Africa (44%) had the lowest adoption rates (table 1, figure 2).

In low-prevalence countries (n=34), 53% of policies (n=18/34) had fully or mostly adopted the guidance and 47% (n=16/34) had generally not adopted guidance. The most common reasons for non-adoption were not using a dual HIV/syphilis RDT in ANC (n=15/34), no retesting prior to ART initiation (n=14/34) and using only two assays to make an HIV-positive diagnosis (n=10/34). In high-prevalence countries (n=11), 91% of policies (n=10/11) had fully or nearly adopted WHO guidance and 9% had not adopted WHO guidance (1/11). The most common reasons for not fully adopting WHO guidance were continuing to use a two-test strategy to make an HIV-positive diagnosis (n=10/11), employing both serial and parallel testing strategies (n=4/11), not adopting dual HIV/syphilis RDTs in ANC (n=4/11) and not implementing retesting prior to ART initiation (n=3/11).

### Adoption of serial testing strategy

In 2021, 62% of national testing strategies (n=29/47) adopted the use of serial testing and 36% (n=16/47) recommended the mix use of serial and parallel testing (figure 3A). The subregions with the highest adoption were Central (78%) and Eastern Africa (75%), followed by Western Africa (38%); Southern Africa had the lowest adoption rate (20%) (table 1). However, fewer countries in Southern Africa recommended serial testing in 2021 (n=1) than in 2018 (n=4) (online supplemental table 4). Most policies recommending serial or parallel testing

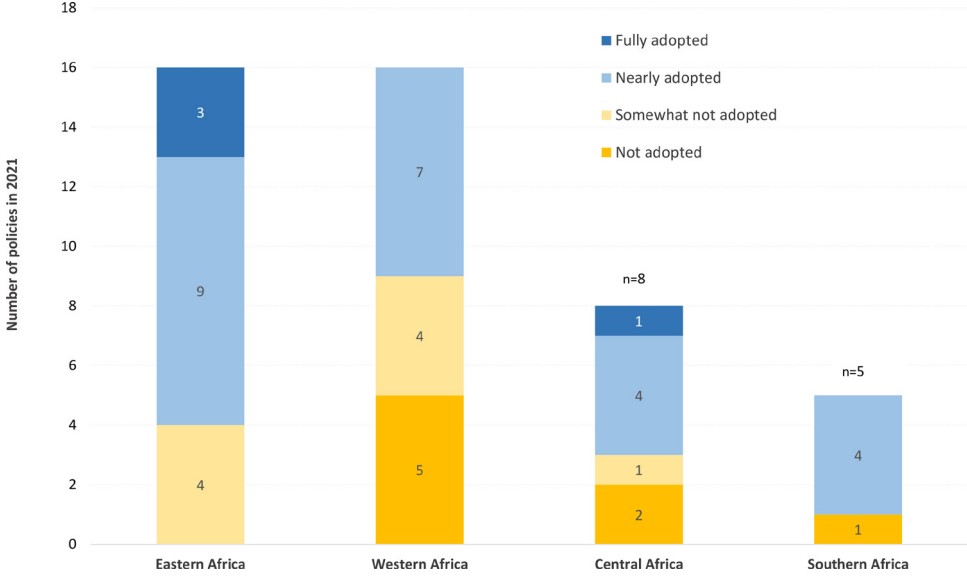

**Figure 2** Overall adoption of 2019 WHO HIV testing strategies in Africa by subregion in 2021.

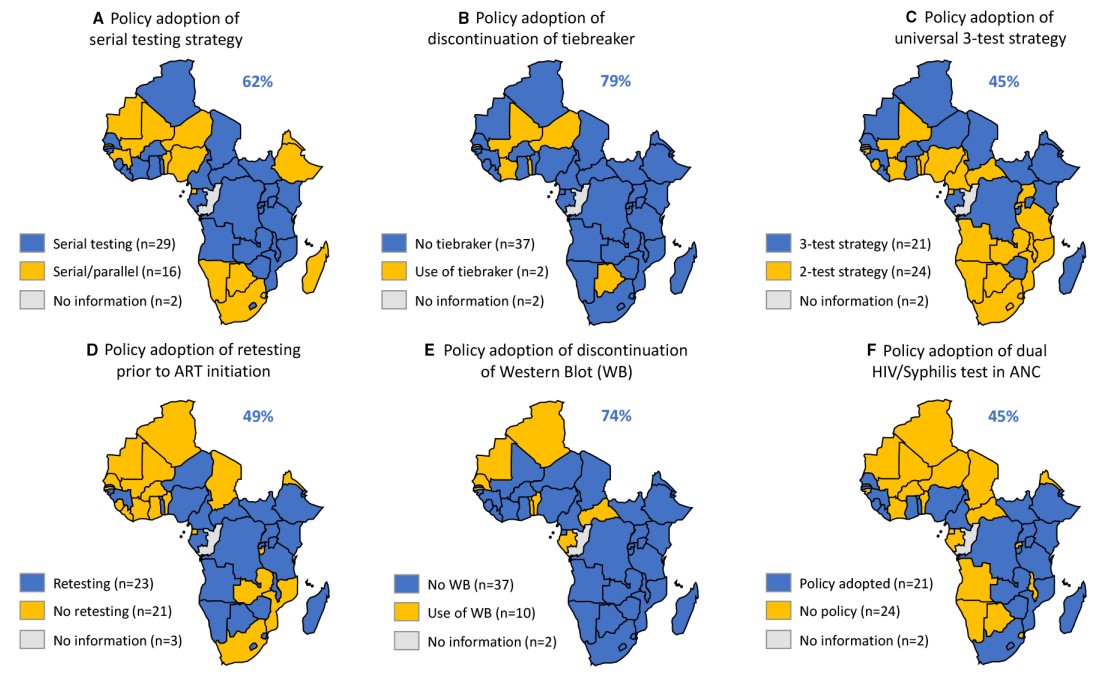

**Figure 3** Policy adoption in 2021 of six specific WHO HIV testing recommendations in the WHO African Region. ANC, antenatal care; ART, antiretroviral treatment.

(n=15/16) conducted simultaneous testing of assay 1 (A1) and assay 2 (A2) in case of discrepant test results, and the Namibian policy recommended parallel testing of A2 and A3 after a reactive A1.

Only 43% of policies (n=20/47) guided the assay order in relation to their sensitivity and specificity. The proportion of HIV testing strategies/algorithms using two consecutive HIV-reactive tests to make an HIV-positive diagnosis was 51% (n=24/47) whereas 45% (n=21/47) recommended the use of three-consecutive HIV-reactive tests.

Sixty-four per cent of the HIV testing strategies reviewed (n=29/45) included information on the testing algorithms, that is, contained specific product names and all of them included WHO prequalified products. Determine HIV 1/2 (Abbott Diagnostics Medical, Japan) was the most common product used as A1 and Bioline HIV 1/2 3.0 (Abbott Diagnostics Korea, Republic of Korea) or Uni-Gold HIV (Trinity Biotech, Ireland) as A2 or A3 (see online supplemental table 5). Only 31% of reviewed policies (n=14/45) mentioned the need to locally verify or validate the HIV testing algorithm.

## Discontinuation of a tiebreaker testing strategy

The discontinuation of a third assay to rule in HIV infection (ie, tiebreaker) was recommended in 78% of national policies (n=37/47) (figure 3B). The subregions with the highest adoption were in Southern Africa and Eastern Africa (100%, each) followed by Central Africa (78%), whereas Western Africa had the lowest adoption (53%) (table 1).

Of the eight policies recommending a tiebreaker testing strategy, seven were from Western Africa and one from Central Africa. Seven policies specified the type

of A3 to be used, for example, RDT (n=4), RDT or EIA (n=1), nucleic acid testing (NAT: n=2) and line immunoassays (LIA (n=1).

## Adoption of WHO standard three-assay testing strategy

The use of three consecutive HIV-reactive tests for the diagnosis of HIV was recommended by 45% of the policies (n=21/47) (figure 3C). The adoption of this WHO recommendation was highest in Eastern Africa (56%) and Western Africa (47%) and lowest in Southern Africa (20%) (table 1). Most policies recommending a three-test strategy (n=20/21) were from low HIV-prevalence countries. Among high HIV-prevalence countries, only 1 out of 11 countries (9%) officially recommended the use of a three-test strategy, which was Zimbabwe.

Overall, across countries that had policies reviewed in both 2018 and 2021, nine policies that previously recommended a two-test strategy transitioned to a three-test strategy as of 2021 (Burkina Faso, Burundi, Chad, Ethiopia, Ghana, Kenya, Lesotho, Madagascar and Zimbabwe, respectively).

Further, we identified that 21% of national policies (n=10) recommended the use of NAT as part of the testing algorithm for the diagnosis of HIV infection in individuals ≥18 months of age. Four policies were from Eastern Africa (Malawi, Mauritius, Mozambique and United Republic of Tanzania); three national policies from Southern Africa (Eswatini, Lesotho and Namibia); two policies from Central Africa (Angola and Sao Tomé et Principe) and one national policy from Western Africa (Guinea Bissau). While not in alignment with WHO guidance, two policies employed NAT as the third assay (A3) of the algorithm while the remaining eight policies recommended NAT to resolve persistent inconclusive/

indeterminate test results (referring either to viral load, PCR or DNA-PCR).

## Adoption of retesting to verify positive status prior to ART initiation

In 2021, 49% of national strategies (n=23/47) recommended retesting prior to ART initiation (figure 3D). The regions with the highest level of policy adoption were in Southern Africa (80%) followed by Central (56%) and Eastern Africa (56%) (table 1). However, fewer countries in Southern Africa recommended retesting before ART initiation in 2021 (n=4) than in 2018 (n=5) (online supplemental table 5). Sixty-five per cent of the retesting policies (n=15/23) recommended retesting using the same national HIV testing algorithm in a new specimen run by a different operator in a different testing site; 26% (n=6/23) did not specify how retesting was carried out and 9% of policies (n=2/23) used retesting strategies that differed from the national HIV testing algorithm, which included parallel testing of A1/A2 in Botswana and Malawi.

## Discontinuation of WB

The proportion of national policies not including WB or LIA as part of the HIV testing algorithm was 74% (n=35/47) (figure 3E). This proportion was highest in Southern Africa (100%) and Eastern Africa (88%) followed by Central Africa (67%) and Western Africa (59%) (table 1). Most countries with a supporting policy in favour of WB or LIA (n=9/10) recommended its use to manage persistent inconclusive/indeterminate test results and one policy used WB as A3 (Mauritius).

## Adoption of dual HIV/syphilis RDT in ANC settings

Forty-five per cent (n=21/47) of policies reviewed supported the use of dual HIV/syphilis RDTs among pregnant women in ANC as recommended by WHO (figure 3F). The region with the highest policy uptake was Southern Africa (60%), followed by Eastern (56%), Western (41%) and Central Africa (22%) (table 1). And of policies supporting a three-test strategy, only 53% (n=11/21) also recommended the use of dual HIV/syphilis RDTs among pregnant women. The majority of policies (n=20/21) recommended the use of dual HIV/syphilis RDT as A1 among pregnant women and only one policy (Lesotho) recommended its use as a triage test (A0) as an interim algorithm while waiting for the results of a national verification study.

The dual HIV/syphilis testing algorithm was not aligned with the national HIV testing algorithm in one country (Zambia): while Determine HIV-1/2 (Abbott Diagnostics Medical, Japan) and SD Bioline HIV-1/2 3.0 (Abbott Diagnostics Korea, Republic of Korea) products were used as A1 and A2 in the national testing algorithm, Bioline HIV/Syphilis Duo (Abbott Diagnostics Korea) and Rapid Test for Antibody to HIV (Colloidal Gold Device) (Beijing Wantai Biological Pharmacy Enterprise, China) were used as A1 and A2 in the ANC testing algorithm.

In addition to the use in ANC settings as recommended by WHO in 2019, three countries also recommended testing with dual HIV/syphilis RDTs among other populations such as male partners of pregnant women and key populations (Liberia, Madagascar and Uganda). Only 50% of policies (n=11/21) specified the brand name of the dual HIV/syphilis RDT in the testing algorithm, and among the policies mentioning the brand name, Bioline HIV/Syphilis Duo (Abbott Diagnostics Korea) was the most employed RDT (100%). Only 57% of policies (n=12/21) mentioned whether syphilis treatment was provided immediately after a reactive syphilis test result.

## DISCUSSION

The current review found a significant improvement in the overall policy uptake of the latest WHO-recommended HIV testing strategies with 60% of countries in the African region adopting WHO guidance in 2021. When directly comparing 32 countries with policies reviewed in 2018 and 2021,[13] policy adoption increased more than 2-fold (20 vs 9). As of the 2021 review, 38% of policies were published before 2019 which underscores the need to update national testing policies at more regular intervals.

Given the rapid changes in HIV testing landscape, greater efforts are needed to support national programmes to update their national HIV testing policies at more regular intervals to keep up with latest WHO guidance. This is particularly critical now as WHO is now updating HTS guidance in 2023-2024 and it will be essential for WHO to continue policy tracking,[10] to allocate sufficient resources, to conduct implementation science research to understand bottlenecks hinder policy change and to support country-led technical working groups to drive policy change.[11 12]

The recommendations with highest adoption were the discontinuation of a tiebreaker (79%) and WB (74%) as well as the use of serial testing (62%). One likely explanation for the higher policy adoption is that these WHO's recommendations were first published between 2012 and 2015, except for the discontinuation of WB which was recommended in 2019. Thus, national HIV programmes have had more time to fully incorporate these recommendations into their national guidelines. Although moving away from WB was recommended more recently,[14] resource-limited countries in the WHO African region have been at the forefront of implementing HIV RDTs compared with other WHO regions as a way to rapidly expand and increase access to HTS.[15]

Our review found that a dozen of countries, mostly in Western Africa, still use WB as part of the HIV testing algorithms, hindering same-day diagnoses and rapid access to ART and PrEP. Additionally, between 2018 and 2021, countries in Southern Africa appeared to be the one subregion which shifted away from serial testing to parallel testing strategies. This shift has likely increased testing costs as it requires more test kits per individual tested. Further

follow-up with countries is needed to understand their policies and should continue to promote serial testing.

Despite observing that many countries recommended retesting prior to ART initiation in 2021, we noted that progress in policy adoption was minimal and that in Southern Africa one less country was implementing when compared with 2018. And only half of countries (49%) had adopted retesting prior to ART initiation in their policies. Retesting prior to ART initiation remains an important quality assurance strategy; however, if countries are struggling to adopt this recommendation it may be important for WHO to provide alternative quality assurance approaches and address implementation challenges.

The WHO's recommendations with the lowest uptake were the use of a standard three-test strategy (45%) and the use of dual HIV/syphilis RDT among pregnant women (45%), both recommendations released in 2019.[6] Before 2019, WHO recommended the use of a two-test strategy in high HIV prevalence settings (≥5%) and the use of a three-test strategy in low HIV prevalence settings (<5%) to maintain at least a 99% positive predictive value.[1] However, as HTS and ART have been scaled up substantially and fewer people undergoing HTS are HIV-positive, the national HTS positivity has also declined, even in high HIV-burden settings.[16] Given these changes in the epidemic, WHO now recommends that all settings move toward using a three-test strategy to ensure high-quality testing.[16]

Transitioning to a three-test strategy should be a priority, particularly for the historically high HIV-burden countries which were using a two-test strategy at the time of this review. This review found that only 9% (n=1/11) of the high-burden countries in the WHO African region have a supportive policy on a three-assay testing strategy. A likely explanation for this slower policy adoption is programmes needing to prioritise limited resources and efforts to address the COVID-19 pandemic. Many programmes, including those focused on HIV testing, were affected by disruptions during COVID-19 and updates and changes to national policies were delayed.[4] Changing a testing algorithm also takes time and planning to implement, including provider training, quality assurance, procurement, inventory management and budget implications to the national programme.[6 16] Many countries indicated they were in the process of transitioning to a three-test strategy but needed to finalise guidance, complete verification studies, conduct tendering process for products, and garner consensus and support among partners. For example, after the review was completed, Malawi updated their guidance and currently recommends a three-test strategy. However, some countries have yet to prioritise adopting the three-test strategy because they were focused on higher yield testing approaches alone,[17 18] or delayed the transition due to costs concerns. While countries will need to develop a plan and identify the optimal time for the transition, in terms of costs, a modelling study found that the total cost of the three-test strategy would have a very limited impact on costs.[19] Further, in Ghana after transitioning to a three-test strategy they found that while there were lessons learnt about logistics and training, the new strategy was found to be feasible, improved testing quality and reduced costs due to preventing misdiagnosis.[20]

We found that among low-prevalence countries (n=34), a significant proportion (41%) still use a suboptimal testing strategy with two assays to make an HIV-positive diagnosis, which will increase the likelihood of false-positive diagnosis.[19–21] A retrospective study in Nigeria using household survey data found that the performance of the two-test strategy in a low prevalence setting of about 1.4% was poor with a PPV of 94% and a false-positive rate of 5.5%.[22] This positive predictive value is similar to a modelling study assessing the accuracy of the 2015 and 2019 WHO HIV testing algorithms, which was 95.4% using a two-assay testing strategy.[19]

Considering that the recommendation to use dual HIV/syphilis RDTs in ANC was released only in 2019,[6 23] its adoption by 21 countries has been encouraging. The number of countries adopting dual HIV/syphilis RDTs among pregnant women continues to increase and may be in practice higher considering that preliminary National Commitments and Policy Instrument (NCPI) survey data reported that a total of 26 countries in the WHO African region adopted the use of dual HIV/syphilis RDTs in ANC.[24] The differences likely reflect distinct methodologies (policy reviews vs policy survey). Often annual global policy surveys, such as NCPI, may reflect a mixture of policies and anecdotal practices, including unofficial policies not yet within published guidelines. Although the findings of this review likely provide a conservative estimate of policy adoption, they do highlight the need to ensure official national policies are updated accordingly.

Some countries may choose not to introduce dual HIV/syphilis RDT for pregnant women. For example, in Malawi where dual HIV/syphilis RDT introduction has been considered but not adopted, the national programme has opted to retain stand-alone HIV and syphilis RDTs in ANC to prioritise point-of-care test options for HIV and syphilis while maintaining a single HIV testing algorithm in a resource-limited context. At the time of this review, three countries also had policies using the dual test outside of pregnant women, including key populations and male partners of pregnant women. As of 2023, WHO now recommends dual HIV/syphilis testing for key populations and those with high ongoing HIV risk and this may further increase its use in HIV testing algorithms in national programmes.[25] It is an important option to increase HIV and syphilis detection and treatment and a recent modelling study in Viet Nam shows that it is cost-effective and cost saving.[26]

As countries update their national HIV testing guidelines to incorporate WHO guidance and adapt their national HIV testing strategies and algorithms to shift towards using three consecutive reactive serology tests to diagnose HIV, introduce dual HIV/syphilis RDTs among pregnant women and key populations, move away from

WB, reinforcing retesting prior to ART initiation as well as design flexible algorithms to address kit shortages, WHO encourages national programmes to conduct a verification study to provide objective evidence, before national scale-up, that a specific combination of products work well together without sharing false-reactivity, thus reducing the risk of misdiagnosis.[6 9] To assist countries, WHO has put together a practical toolkit to accelerate policy adoption.[27] To date, the following African countries have worked directly with WHO to update their testing algorithm: Burkina Faso, Cameroon, Chad, Central African Republic, Cote D'Ivoire, Democratic Republic of Congo, eSwatini, Guinea, Kenya, Lesotho, Malawi, Mali, South Sudan and Zambia.

This policy review has strengths as well as weaknesses. First, this review provides a comprehensive review of existing national HTS policies in the African region and was able to identify current policies for 45 countries. While this represents nearly all countries in the region, we were unable to identify policies for two countries. Second, unlike other policy surveys, this study reviewed official policies for each country that were collected through a robust global search and triangulated data with additional information provided by country-level key informants. Despite this, the review was unable to assess unpublished policies and those in development at the time of the analysis. During the review, some countries indicated that they were in the process of updating their guidance, particularly around adoption of a three-test strategy. We were unable to fully include this information in the review, and thus, our reported level of adoption of WHO recommendations may be an underestimate. Third, this review did not assess policy implementation at the national, subnational or site level. Thus, some practices in the country may differ than policies reviewed, and level of adoption reported may vary. Lastly, we focused on HIV testing strategies for individuals ≥18 months of age. As a result, we were unable to report on the adoption of HIV testing strategy recommendations for children, such as early infant diagnosis.

## CONCLUSIONS

Adoption of WHO-recommended HIV testing strategies has improved in the African region. While WB was only used in a few countries, concerted efforts are still needed to phase out this technology in favour of RDTs. Countries should plan to accelerate their transition to WHO recommendations by streamlining efforts to adopt and implement a three-asssay testing strategy, retesting before ART initiation and use of dual HIV/syphilis RDTs. Conducting verification of testing algorithms and using appropriate assays can ensure accurate HIV diagnosis in a cost-efficient and time-efficient manner. Greater efforts are needed now to ensure countries implement high-quality testing services as they are essential for the global goal to achieve and maintain low HIV incidence.

**Author affiliations**
[1]Global HIV, Hepatitis, and STI Programmes, World Health Organization, Geneva, Switzerland
[2]World Health Organization Regional Office for Africa, Bamako, Mali
[3]Inter-country support team for Central Africa, World Health Organization Regional Office for Africa, Libreville, Gabon
[4]Inter-country support team for Western and Central Africa, World Health Organization Regional Office for Africa, Ouagadougou, Burkina Faso
[5]Inter-country support team for Eastern and Southern Africa, World Health Organization Regional Office for Africa, Harare, Zimbabwe
[6]World Health Organization Regional Office for Africa, Brazzaville, Congo
[7]Regulation and Prequalification, World Health Organization, Geneva, Switzerland

**Acknowledgements** We would like to express our gratitude to WHO colleagues in the respective countries of the WHO African region for their support in locating updated national policy documents. We would also like to thank staff at the Global Fund (Youssouf Sawadogo, Ghislaine Grasser and Jacqueline Papo), CHAI (Katherine Guerra, Gillian Leitch, Megan Ginivan and Christian Stillson), Evidence Action Group (Anna Konstantinova and Emilie Efroson) and staff from Ministry of Health in Mozambique (Helga Guambe) and Guinea Bissau (David da Silva Té) for their support in identifying latest national HIV testing policies.

**Contributors** CJ and AS devised and CJ and RB supervised the review. CJ and RB were responsible for the study. EF led the study protocol developments, conducted the screening, data extraction and analysis. NB, CMM, NSB, FM, FJL, MASP and LH assisted with acquisition of policies, data collection and results interpretation. CL acted as a second reviewer and CJ as third reviewer. AS, MSJ, MB-D and RB assisted with assessing policies and interpreting results. EF led manuscript writing along with CJ, with support from all authors. CL, NB, CMM, NSB, FM, FJL, MASP, LH, MB-D, MSJ, AS and RB all reviewed the first draft, provided critical review and input and approved the final version of the manuscript.

**Funding** This work was supported by the Bill & Melinda Gates Foundation, grant number INV-024432.

**Disclaimer** The author is a staff member of the World Health Organization. The author alone is responsible for the views expressed in this publication and they do not necessarily represent the views, decisions or policies of the World Health Organization. The funders had no role in study design, data collection and analysis, decision to publish, or preparation of the manuscript. The named authors alone are responsible for the views expressed in this publication.

**Map disclaimer** The inclusion of any map (including the depiction of any boundaries therein), or of any geographic or locational reference, does not imply the expression of any opinion whatsoever on the part of BMJ concerning the legal status of any country, territory, jurisdiction or area or of its authorities. Any such expression remains solely that of the relevant source and is not endorsed by BMJ. Maps are provided without any warranty of any kind, either express or implied.

**Competing interests** None declared.

**Patient and public involvement** Patients and/or the public were not involved in the design, or conduct, or reporting, or dissemination plans of this research.

**Patient consent for publication** Not applicable.

**Provenance and peer review** Not commissioned; externally peer reviewed.

**Data availability statement** Data are available on reasonable request. All data relevant to the study are included in the article or uploaded as online supplemental information. All data generated in the study are included in the article or uploaded as online supplemental information. National policies may be publicly available. Some policies included in this review may be available through the following websites: (1) https://aidsfree.usaid.gov/resources/guidance-data/hts. (2) http://www.hivpolicywatch.org/database.html. If information on a policy cannot be found through these resources, please contact the authors of this review for additional information.

for any error and/or omissions arising from translation and adaptation or otherwise.

**ORCID iD**
Emmanuel Fajardo http://orcid.org/0000-0002-7561-3368

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
