## [Reviewer comments · BMJ Open]

ARTICLE DETAILS

TITLE (PROVISIONAL)	Country adoption of WHO 2019 guidance on HIV testing strategies and algorithms: a policy review across the WHO African region
AUTHORS	Fajardo, Emmanuel; Lastrucci, Céline; Bah, Nayé; Mingiedi, Casimir Manzeno; Ba, Ndoungou Salla; Mosha, Fausta; Lule, Frank John; Paul, Margaret Alia Sampson; Hughes, Lago; Barr-DiChiara, Magdalena; Jamil, Muhammad; Sands, Anita; Baggaley, R; Johnson, Cheryl

VERSION 1 – REVIEW

REVIEWER	Domingues, Rosa INI/Fiocruz, LAPCLIN AIDS
REVIEW RETURNED	19-Apr-2023

GENERAL COMMENTS	The paper is relevant considering the importance of HIV testing for the prevention and control of the disease and the governmental adoption of WHO recommendations as an essential step for the implementation of guidelines. However, some aspects are not clear and could be improved: 1) Authors should inform more clearly the assessment that was made. Only in the results section it became clear to me that of the six recommendations that were evaluated, two had already been evaluated in 2014 and 2018, one in 2018 and three only in 2021; 2) I have questions related to the overall adoption assessment: a) The authors make a global assessment of the recommendation adoption and compare the results observed in 2021 with those observed in previous periods. However, it does not seem appropriate to make this comparison. Firstly, because the countries that were subject to assessment are not the same. Did the countries included in 2021 already have high adoption in 2018? If so, the better observed results could be attributed to the inclusion of these countries and not to an increase in adoption rates. In addition, the evaluated recommendations are not the same. Could differences in recommendations alone explain the higher adoption in 2021? b) The authors did not inform the reference used to assess global adoption - adopted (6), nearly adopted (5 to 4), somewhat not adopted (3), not adopted (2 or less) - nor the justification for grouping this categories into two groups (lines 157-158 and lines 161-162). Why classify into four categories if they will be grouped into two? With this classification, countries with 4 or 6 adopted recommendations are classified as “adopted”, which does not reflect the same level of adoption. For example, in Western and Southern Africa regions, no policy had full adoption of recommendations.
--

	3) While global adoption has improved, adoption of two recommendations has declined over the periods evaluated and these results were not discussed by the authors: serial testing strategy decreased in all regions and retesting prior to ART initiation in two regions. 4) It is not clear how the authors assessed the recommendation “universal 3-test strategy”. In lines 113-116 the authors inform that “However, as ART coverage expands and HIV positivity declines, WHO now recommends countries to use the national HTS positivity and the treatment-adjusted prevalence, which accounts for both the national HIV prevalence and ART coverage, as an indicator to determine when to transition to a three-assay testing strategy”. Afterwards, in the discussion section, the authors say that “Given this changes in the epidemic, WHO now recommends that all settings move toward using a 3-assay testing strategy to ensure high-quality testing. And in another part of the discussion section that “Specifically, in high HIV burden settings in southern Africa (5 countries) and eastern Africa (6 countries), WHO recommends countries still using a 2-assay testing strategy to move toward using a 3-assay testing strategy as their treatment-adjusted prevalence, rather than the national HIV prevalence, and national HTS positivity fall below 5%”. What was considered in the evaluation of these countries to classify the recommendation as adopted? Has an individual assessment been made of each country to see if it should already be using the three-trial testing strategy? 5) Use of nucleic acid testing (NAT) to diagnose HIV infection was described in the results (as a topic) but it is not part of the recommendations to be verified. Please review. Minor corrections: Line 102: please include number “1” into parentheses Lines 172-173: The numbers are not clear. The authors inform that they identified updated policies in 41 countries. Then they inform that they used policies identified in 2018 for 6 countries, but the total was 45/47. Please clarify. Line 263: please review the sentence “The region was the highest policy uptake was Southern...” Line 435: please remove the parentheses after number 99 The authors informed the categories for data extraction in supplementary table 1. I think it would be useful for readers who are not familiar with new recommendations to present them in a box, with a brief description of each one of them. This information should help spread the WHO recommendations to those who are not aware of them.
--	--

REVIEWER	Puttkammer, Nancy University of Washington School of Public Health, I-TECH, Department of Global Health
REVIEW RETURNED	01-May-2023

GENERAL COMMENTS	This paper presents results of a review of HIV testing policies in 45 African countries, following the WHO’s 2019 HTS guidelines. It is a very useful contribution to understand how policy adoption has proceeded with respect to these guidelines. Overall comments o Major comment: I am concerned with the presentation of the results on longitudinal comparison of guideline compliance. It is problematic to make the claim of 2.4 fold improvement in policy
---

	compliance given that only 32 policies were reviewed in 2018 compared to 45 in 2021. Those not reviewed (unobserved) in 2018 could represent significant bias in the results. For example, if all 13 of the unobserved cases were actually compliant with adoption in 2018, this would have been a compliance level of $n=22/32=69\%$ -- similar to the level observed in 2021. It would be a fairer comparison longitudinally to compare the same 32 countries. o For readers who are less familiar with the criteria for moving from the 2-test to the 3-test strategy, the paper might benefit from presenting this information a bit differently. Lines 114-116 state: "However, as ART coverage expands and HIV positivity declines, WHO now recommends countries to use the national HTS positivity and the treatment-adjusted prevalence, which accounts for both the national HIV prevalence and ART coverage, as an indicator to determine when to transition to a three-assay testing strategy". This sentence suggests that some countries should retain the two-assay testing strategy. However, the review paper seems to consider use of the two-assay strategy as non-compliant with the 2019 guideline. If there is some nuance on whether or not all countries should transition, perhaps this should not be considered as a criteria for adoption vs. non-adoption of the 2019 guidelines. However, the authors could still provide descriptive results on where guidelines stand as of 2021 with respect to this issue. o The authors are inconsistent in labeling and presenting the key guideline policy recommendations that are included in the 2019 HTS guidance. I recommend aligning the phrasing used and the order in which the key policy recommendations are presented and discussed across the paper. For example, the areas of guideline compliance as listed in the abstract do not fully align with the results in Table 1 or in the subsections shown in the results section, in terms of order in which the items are shown. Lines 314-323 adds a sub-section on NAT, but this does not appear in Table 1 and the manuscript is not clear on the 2019 WHO guidelines with respect to NAT. Can this be folded into other sub-sections so that the results section stays aligned with the 6 areas of guidelines? o I would like to see the authors situate their inquiry in the broader base of knowledge about effective policy dissemination and implementation. There is a body of literature on policy implementation science, guideline dissemination and implementation, and policy change process that would be very interesting to reference and discuss as part of the paper. I recommend that the background/introduction section include 2-3 sentences about how the 2019 WHO HTS guidelines were disseminated to countries and globally. Then, in the Discussion section, it would be helpful to situate the findings in terms the policy change processes in countries. Policy change is not easy, and requires multiple steps and stakeholder engagement in each country setting. The authors spend most of the discussion section focused on very specific issues about HTS without zooming out to discuss their recommendation that countries should update their guidelines and algorithms more rapidly or frequently. It would be helpful if they can put this recommendation in the context of acknowledging what it takes to make policy changes. The discussion of algorithm validation would fit well in this part of the discussion, recognizing that many countries feel a need to validate WHO's global recommendations in their own contexts, or at least go through a formal policy review and adoption process that considers if/how to adapt WHO guidelines to their settings. The
--	--

following papers might provide useful grounding in policy implementation concepts and principles:

- o Nilsen P, Ståhl C, Roback K, Cairney P. Never the twain shall meet?--a comparison of implementation science and policy implementation research. *Implement Sci.* 2013 Jun 10;8:63. doi: 10.1186/1748-5908-8-63. PMID: 23758952; PMCID: PMC3686664.
- o Oh A, Abazeed A, Chambers DA. Policy Implementation Science to Advance Population Health: The Potential for Learning Health Policy Systems. *Front Public Health.* 2021 Jun 17;9:681602. doi: 10.3389/fpubh.2021.681602. PMID: 34222180; PMCID: PMC8247928.
- o Flodgren G, Hall AM, Goulding L, Eccles MP, Grimshaw JM, Leng GC, Shepperd S. Tools developed and disseminated by guideline producers to promote the uptake of their guidelines. *Cochrane Database Syst Rev.* 2016 Aug 22;(8):CD010669. doi: 10.1002/14651858.CD010669.pub2. PMID: 27546228.

- The writing is a bit awkward in spots. Common typos include incorrect verb tense and other. I recommend further copy editing for concise and clear writing. For example,
 - o Line 264-66 is presently written as: “Among high HIV-burden countries, only 18% of policies (n=2/11) recommended the use of a 3-assay testing strategy and this corresponded to two countries, Malawi and Zimbabwe, respectively.” It would be clearer and simpler to write “Only two out of eleven high HIV-burden countries, Malawi and Zimbabwe, recommended the use of a 3-assay testing strategy (18%).”
 - o Line 277-279: “Sixty-five percent of the policies (n=15/23) recommending retesting prior to ART initiation, conduct retesting using the same national HIV testing algorithm in a new specimen run by a different operator in a different testing site” could be rewritten as “Sixty-five percent of the retesting policies (n=15/23) recommended retesting using the same national HIV testing algorithm in a new specimen run by a different operator in a different testing site”.

Specific comments

Abstract

- The n’s used in the Results paragraph are a bit confusing. Sentence on lines 43-45 is not clear. “Using a 2-assay testing strategy was the most common form of non-adoption: 31% (n=14) and 22% (n=10).” It is not immediately clear if the n’s shown there represent numerators or denominators.
- In the Abstract, I recommend focusing more on the descriptive results in 2021 rather than the longitudinal change in guidelines adoption from 2018 to 2021 (see reasons above). However, if the authors wish to present a longitudinal comparison for the same countries assessed at both time points, as part of the Abstract, this would be OK at the end of the results section of the Abstract.

Introduction

- In the paragraph starting on line 97, it would help to underscore the importance of the 2019 guideline changes if the authors could summarize in 2-3 sentences the advantages of the guideline changes. Does the evidence base for the changes demonstrate improved (or at least non-inferior accuracy), less drop-out in the HIV cascade, lower cost, greater feasibility or simplicity of HTS services, etc?

Methods

	 • Line 148-49: If something was marked as ‘unclear’ within the documents gathered, was further information sought from key informants to clarify? Results  • Line 193: See comment above about longitudinal comparison. It is not useful to contain the absolute numbers or relative increase in policies adopting WHO HTS guidelines, because the denominators were different in 2018 vs. 2021. • Line 270: word “respectively” is not needed. • Line 287: typo “Easter Africa” • Line 271-272 references guidelines on dual HIV/syphilis testing, and it seems this would be better placed in the subsection on that topic (line 291) • Line 306: can you name the three countries recommending dual HIV/syphilis testing among groups outside of ANC? Discussion  o Line 330: I think it is misleading to start the sentence with “However”. Your point should be that “as of the 2021 review, 40% of policies were published before 2019 which underscores the need to update national testing policies at more regular intervals.” o Line 339: typo “WHO’s” should be “WHO” o Line 343: “has” should be “have” (countries is plural) o Line 361: “given this changes” should be “given these changes” o Line 399: would be helpful to define the acronym NCPI and explain what is meant by this. The following sentence is unclear on the authors’ interpretation of possible bias. Are they saying that policy reviews are never to be trusted? Or that their review is accurate while surveys of policy are not? o Lines 414-416 are unnecessary in the sentence. Overall the paragraph on conducting algorithm verification studies could be shortened, as it is not the central focus of this paper. o I recommend more discussion of the strengths and limitations in the body of the paper. The authors include bullet points but do not include these in the body of the Discussion section of the paper. Figures  o Figures 2 and 3 are helpful, but could be difficult for readers with red-green color blindness to interpret. Please consider using a different color contrast.
--	---

VERSION 1 – AUTHOR RESPONSE

Reviewer: 1 Dr. Rosa Domingues, INI/Fiocruz

Comments to the Author: The paper is relevant considering the importance of HIV testing for the prevention and control of the disease and the governmental adoption of WHO recommendations as an essential step for the implementation of guidelines. However, some aspects are not clear and could be improved:

- 1) Authors should inform more clearly the assessment that was made. Only in the results section it became clear to me that of the six recommendations that were evaluated, two had already been evaluated in 2014 and 2018, one in 2018 and three only in 2021;

All six recommendations were evaluated in this review in 2021. We have made this clearer in the abstract, methods. The review follows on from those conducted in 2014 and 2018 and looked for changes in policies as well as align with new 2019 WHO recommendations.

See revisions on lines 159-179:

We assessed national adoption of WHO HTS guidance using six specific recommendations related to testing strategies and algorithms for those ≥18 months of age set forth in the WHO 2019 HIV testing guidelines, namely: (1) use of serial testing, (2) use of a 3-test strategy, (3) discontinuation of a tie-breaker to rule in HIV infection, (4) discontinuation of WB/LIA, (5) retesting prior to ART initiation, and (6) use of dual HIV/syphilis RDT in ANC. Additional qualitative details about the national testing algorithm, such as the order of and name brand of test kits used in testing algorithms, were also reviewed when reported and assessed according to WHO recommendations (see Box 1).

Based on the number of recommendations adopted, national policies were categorized as: adopted (6); nearly adopted (5 to 4); somewhat not adopted: (3); and not adopted: (2 or less). We then provide simplified reporting (adopted, partially adopted and not adopted) for national policies which reviewed in 2018 and 2021 to assess changes overtime. Notably, these categories were initially developed as part of the 2018 policy review and maintained to assist with future updates and policy tracking.

Descriptive analyses were then stratified by subregions (Western, Central, Eastern and Southern Africa; online supplementary figure 1) were also conducted to determine rates of adherence by subregion. All analyses were conducted in Microsoft Excel. Countries which had policies reviewed in 2018 and 2021 were also compared to assess changes in alignment with WHO recommendations over time.

See also the additional box summarising the guidance assessed as part of the review starting on line 110:

Box 1. WHO recommendations for HIV testing strategies for individuals ≥18 months of age, 2019

- *Western blotting and line immunoassays should not be used in national HIV testing strategies and algorithms.*
- *Dual HIV/syphilis RDTs can be the first test in HIV testing strategies and algorithms in ANC settings.*
- *In response to changes in the HIV epidemic, WHO recommends countries use three consecutive reactive tests to provide an HIV-positive diagnosis.*
- *Testing strategies should use tests serially, not in parallel, and should not use a tiebreaker to rule-in HIV infection. Instead, those with discrepant test results should be ruled inconclusive and referred for further testing in 14 days to rule-in or rule-out seroconversion.*
- *WHO recommends that all HIV testing algorithms use a combination of RDTs and/or EIAs to achieve at least 99% positive predictive value and use a combination of tests with ≥99% sensitivity and ≥98% specificity. WHO does not recommend the use of NAT techniques within HIV testing algorithms for individuals ≥18 months of age.*
- *The first test in an HIV testing strategy and algorithm should have the highest sensitivity, followed by a second and third test of the highest specificity. Algorithms should be validated and verified to ensure high quality and accurate testing services.*
- *All people newly diagnosed with HIV should be retested to verify their HIV status prior to starting ART, using the same testing strategy and algorithm as the initial test. To minimize the risk of misdiagnosis, this approach should be maintained in settings in which rapid ART initiation is being implemented.*

Source: WHO 2019, [6]

- 2) I have questions related to the overall adoption assessment: a) The authors make a global assessment of the recommendation adoption and compare the results observed in 2021 with those observed in previous periods. However, it does not seem appropriate to make this comparison. Firstly, because the countries that were subject to assessment are not the same.

Did the countries included in 2021 already have high adoption in 2018? If so, the better observed results could be attributed to the inclusion of these countries and not to an increase in adoption rates. In addition, the evaluated recommendations are not the same. Could differences in recommendations alone explain the higher adoption in 2021? b) The authors did not inform the reference used to assess global adoption - adopted (6), nearly adopted (5 to 4), somewhat not adopted (3), not adopted (2 or less) - nor the justification for grouping this categories into two groups (lines 157-158 and lines 161-162). Why classify into four categories if they will be grouped into two? With this classification, countries with 4 or 6 adopted recommendations are classified as “adopted”, which does not reflect the same level of adoption. For example, in Western and Southern Africa regions, no policy had full adoption of recommendations.

We have made it clearer that this review assessed policies in all countries. It is a follow-on study from 2014 and 2018 and the 32 countries with policies that were reviewed in 2018 and 2021. We have also addressed the language throughout the manuscript and now only report adoption overall and do not directly report an increase.

See revisions on lines 186-187:

In total, we were able to identify policy documents from 96% of countries from the WHO African region (n=45/47); 71% of countries had a policy reviewed in both 2018 and 2021 (n=32/45).

See revisions on lines 206-211:

When the analysis was restricted to a subset of countries with policies in both the 2018 and 2021 review (n=32), the proportion adopting WHO guidance in 2018 was 28% (n=9/32) vs 62% (n=20/32) in the current (2021) review (see online supplementary table 3; supplementary figure 1). Across the sub-regions, Southern (80%) and Eastern (75%) Africa had the greatest level of alignment, whilst Central Africa (63%) and Western Africa (44%) had the lowest adoption rates (table 1; figure 2).

In terms of the categorisations, we utilised a protocol from the 2014 and 2018 reviews (<https://gh.bmj.com/content/5/5/e001939>) which developed the criteria and initial categories. We maintained these categories to ensure greater consistency of the assessment overtime and report them in Figure 2 (revised attached and below). However, we have improved the discussion of the categories and refer now to the status of fully adopted, partially adopted and not adopted policies when we compare policies reviewed in 2018 and 2021, in addition to the 4-categories, to address your concern.

See revision on lines 121-128:

A global review of national HIV testing policies was conducted by WHO in 2018 to assess adoption of its testing recommendations and policies.[13] Of 91 policies reviewed only 24 (25%) adhered to WHO guidance. This policy review follows on from the previous review and seeks to assess country uptake of six 2019 WHO recommendations related to HIV testing strategies. Here we focus on the WHO African region, which hosts several countries with a high HIV burden and with large HIV testing programmes. The same policy review across the remaining five WHO regions is underway as part of the update to the 2023 WHO HTS guidelines.

See revision on lines 141-142:

The full protocol for the review was previously developed and published in 2018. [13]

See revision on lines 168-173:

Based on the number of recommendations adopted, national policies were categorized as: adopted (6); nearly adopted (5 to 4); somewhat not adopted: (3); and not adopted: (2 or less). We then provide simplified reporting (adopted, partially adopted and not adopted) for national policies which reviewed in 2018 and 2021 to assess changes overtime. Notably, these categories were initially developed as part of the 2018 policy review and maintained to assist with future updates and policy tracking.

See revision on lines 204-211:

In 2021, 62% of national testing strategies (n=28/45) were either fully or partially aligned with WHO 2019 HTS recommendations and 38% (n=17/45) had not adopted the recommendations (table 1). When the analysis was restricted to a subset of countries with policies in both the 2018 and 2021 review (n=32), the proportion adopting WHO guidance in 2018 was 28% (n=9/32) vs 62% (n=20/32) in the current (2021) review (see online supplementary table 3; supplementary figure 1). Across the sub-regions, Southern (80%) and Eastern (75%) Africa had the greatest level of alignment, whilst Central Africa (63%) and Western Africa (44%) had the lowest adoption rates (table 1; figure 2).

See supplementary table 3 for the additional reporting on policy adoption:

Supplementary Table 3. Progress on policy adoption in the WHO African Region from 2018 to 2021

Subregion	Count	Country	Policy reviewed in 2018?	Adherence in 2018	Policy reviewed in 2021	Adherence in 2021
Central Africa	1	Angola	Yes, 2015	Not adopted	2020	Mostly adopted
	2	Cameroon	Yes, 2015	Not adopted	2019	Mostly adopted
	3	CAR	Yes, 2010	Not adopted	2018	Not adopted
	4	Chad	Yes, 2011	Not adopted	2017	Mostly adopted
	5	DRC	Yes, 2017	Adopted	2020	Adopted
	6	Equatorial Guinea	No	–	2018	Not adopted
	7	Gabon	No	–	2017	Mostly adopted
	8	Republic of the Congo	No	–	–	–
	9	São Tomé e Príncipe	No	–	2018	Not adopted
Eastern Africa	10	Burundi	Yes, 2016	Not adopted	2020	Mostly adopted
	11	Comoros	Yes, 2007	No information	2016	Not adopted
	12	Eritrea	No	–	2019	Not adopted
	13	Ethiopia	Yes, 2017	No information	2018	Mostly adopted
	14	Kenya	Yes, 2017	Mostly adopted	2021	Adopted
	15	Madagascar	Yes, 2011	No information	2018	Mostly adopted
	16	Malawi	Yes, 2016	Not adopted	2016	Mostly adopted
	17	Mauritius	No	–	2020	Not adopted
	18	Mozambique	Yes, 2016	Not adopted	2020	Mostly adopted
	19	Rwanda	Yes, 2016	Not adopted	2018	Mostly adopted
	20	Seychelles	No	–	2019	Not adopted
	21	South Sudan	Yes, 2017	Not adopted	2020	Adopted
	22	Tanzania	Yes, 2017	Not adopted	2021	Mostly adopted
	23	Uganda	Yes, 2016	Mostly adopted	2020	Mostly adopted
	24	Zambia	Yes, 2018	Not adopted	2020	Not adopted
25	Zimbabwe	Yes, 2016	Adopted	2016	Adopted	
Southern Africa	26	Botswana	Yes, 2016	Mostly adopted	2016	Not adopted
	27	eSwatini	Yes, 2018	Not adopted	2018	Not adopted
	28	Lesotho	Yes, 2016	Adopted	2021	Mostly adopted
	29	Namibia	Yes, 2016	No information	2018	Not adopted
	30	South Africa	Yes, 2016	Mostly adopted	2020	Not adopted

Western Africa	31	Algeria	Yes, 2013	Adopted	2013	Not adopted
	32	Benin	No	–	2017	Not adopted
	33	Burkina Faso	Yes, 2008	Not adopted	2021	Mostly adopted
	34	Cabo Verde	No	–	–	–
	35	Ghana	Yes, 2014	Not adopted	2019	Mostly adopted
	36	Guinea	No	–	2019	Mostly adopted
	37	Guinea-Bissau	No	–	2021	Not adopted
	38	Ivory Coast	Yes, 2016	Not adopted	2016	Not adopted
	39	Liberia	Yes, 2015	Not adopted	2020	Mostly adopted
	40	Mauritania	No	–	2020	Not adopted
	41	Mali	No	–	2017	Not adopted
	42	Niger	No	–	2020	Not adopted
	43	Nigeria	Yes, 2016	Not adopted	2020	Mostly adopted
	44	Senegal	Yes, 2017	Mostly adopted	2018	Not adopted
	45	Sierra Leone	Yes, 2017	Not adopted	2020	Not adopted
	46	The Gambia	Yes, 2014	Not adopted	2019	Not adopted
	47	Togo	No	–	2019	Not adherent

A total of 32 countries had policies reviewed in 2018 and 2021. Those highlighted in red are policies reviewed in 2018 that were updated (n=28). Number of policies deemed adopted in 2018 (n=7) and number deemed adopted in 2021 (n=20)

- 3) While global adoption has improved, adoption of two recommendations has declined over the periods evaluated and these results were not discussed by the authors: serial testing strategy decreased in all regions and retesting prior to ART initiation in two regions.

Thank you for this. We have addressed this by adding in specific points in the results and discussion.

Lines 241-242:

However, fewer countries in southern Africa recommended serial testing in 2021 (n=1) than in 2018 (n=4) (supplementary table 5).

Lines 287-288:

However, fewer countries in southern Africa recommended retesting before ART initiation in 2021 (n=4) than in 2018 (n=5) (supplementary table 5).

Line 371-375:

Additionally, between 2018 and 2021, countries in southern Africa appeared to be the one sub-region which shifted away from serial testing to parallel testing strategies. This shift has likely increased testing costs as it requires more test kits per individual tested. Further follow-up with countries is needed to understand their policies and should continue to promote serial testing.

Line 377-382:

Despite observing that many countries recommended retesting prior to ART initiation in 2021, we noted that progress in policy adoption was minimal and that in southern Africa one less country was implementing when compared to 2018. Retesting prior to ART initiation remains an important quality assurance strategy, however if countries are struggling to adopt this recommendation it may be important for WHO to provide alternative approaches and address implementation challenges.

- 4) It is not clear how the authors assessed the recommendation “universal 3-test strategy”. In lines 113-116 the authors inform that “However, as ART coverage expands and HIV positivity declines, WHO now recommends countries to use the national HTS positivity and the treatment-adjusted prevalence, which accounts for both the national HIV prevalence and ART

coverage, as an indicator to determine when to transition to a three-assay testing strategy". Afterwards, in the discussion section, the authors say that "Given this changes in the epidemic, WHO now recommends that all settings move toward using a 3-assay testing strategy to ensure high-quality testing. And in another part of the discussion section that "Specifically, in high HIV burden settings in southern Africa (5 countries) and eastern Africa (6 countries), WHO recommends countries still using a 2-assay testing strategy to move toward using a 3-assay testing strategy as their treatment-adjusted prevalence, rather than the national HIV prevalence, and national HTS positivity fall below 5%". What was considered in the evaluation of these countries to classify the recommendation as adopted? Has an individual assessment been made of each country to see if it should already be using the three-trial testing strategy?

We have addressed this by clarifying that WHO recommends all countries use a 3-test strategy in response to global trends and epidemic shifts. This guidance is referenced in 2019 and there is no need to assess individual countries to determine which should transition.

See revision on lines 99-109:

Additionally, many countries still use the 2015 WHO recommendation of using the national HIV prevalence to determine whether a two-assay ($\geq 5\%$) or three-assay testing strategy ($< 5\%$) should be used.[1] However, as ART coverage has expanded HIV positivity and the proportion of people with HIV who are undiagnosed and not in care will continue to decline. As a result, the positive predictive value of previous testing strategies will also decline and lead to an increase in false positive diagnoses. In relation to the guideline development process, WHO now recommends all countries use a standard 3-test strategy.[8] WHO also recommends that countries planning to update their HIV testing algorithms undertake a verification study to select appropriate HIV serology products and ensure they don't cross-react in order to minimize the risk of misdiagnosis.[9]

See revision on lines 121-128:

A global review of national HIV testing policies was conducted by WHO in 2018 to assess adoption of its testing recommendations and policies.[10] Of 91 policies reviewed only 24 (25%) adhered to WHO guidance. This policy review follows on from the previous review and seeks to assess country uptake of six 2019 WHO recommendations related to HIV testing strategies. Here we focus on the WHO African region, which hosts several countries with a high HIV burden and with large HIV testing programmes. The same policy review across the remaining five WHO regions is underway as part of the update to the 2023 WHO HTS guidelines.

- 5) Use of nucleic acid testing (NAT) to diagnose HIV infection was described in the results (as a topic) but it is not part of the recommendations to be verified. Please review. Minor corrections: Line 102: please include number "1" into parentheses Lines 172-173: The numbers are not clear. The authors inform that they identified updated policies in 41 countries. Then they inform that they used policies identified in 2018 for 6 countries, but the total was 45/47. Please clarify. Line 263: please review the sentence "The region was the highest policy uptake was Southern..." Line 435: please remove the parentheses after number 99 The authors informed the categories for data extraction in supplementary table 1. I think it would be useful for readers who are not familiar with new recommendations to present them in a box, with a brief description of each one of them. This information should help spread the WHO recommendations to those who are not aware of them.

Thank you for these comments and suggestion to add a box detailing WHO recommendations (see above Box 1). We have addressed the minor corrections as well in the revisions below.

In terms of the use of NAT, WHO does not recommend the use of NAT for those over 18 months of age and therefore a fully summary on the use of NAT in testing policies is outside

the scope of this review which specifically on testing strategies for those ≥ 18 months of age. We have now clarified this in the abstract and methods. Because of growing interest in this area, we have kept the synthesis of policies indicating NAT is used within testing strategies among adults when reported.

See revision on lines 28-29:

Design: A comprehensive review and analysis of HIV testing strategies for individuals ≥ 18 months of age.

See revision on lines 159-166:

We assessed national adoption of WHO HTS guidance using six specific recommendations related to testing strategies and algorithms for those ≥ 18 months of age set forth in the WHO 2019 HIV testing guidelines, namely: (1) use of serial testing, (2) use of a 3-test strategy, (3) discontinuation of a tie-breaker to rule in HIV infection, (4) discontinuation of WB/LIA, (5) retesting prior to ART initiation, and (6) use of dual HIV/syphilis RDT in ANC. Additional qualitative details about the national testing algorithm, such as the order of and name brand of test kits used in testing algorithms, were also reviewed when reported and assessed according to WHO recommendations (see Box 1).

See revision in lines 86-92:

... three new recommendations related to testing strategies namely: (1) moving toward a universal 3-assay testing strategy using three consecutive HIV-reactive tests to provide and HIV-positive diagnosis, (2) discontinuation of western blot (WB) and line immunoassays (LIA) and, (3) use of dual HIV/syphilis RDTs as first assay in HTS in antenatal care (ANC) settings.

See revision in box after line 110:

Box 1. WHO recommendations for HIV testing strategies, 2019

- *Box 1. WHO recommendations for HIV testing strategies for individuals ≥ 18 months of age, 2019*
- *Western blotting and line immunoassays should not be used in national HIV testing strategies and algorithms.*
- *Dual HIV/syphilis RDTs can be the first test in HIV testing strategies and algorithms in ANC settings.*
- *In response to changes in the HIV epidemic, WHO recommends countries use three consecutive reactive tests to provide an HIV-positive diagnosis.*
- *Testing strategies should use tests serially, not in parallel, and should not use a tiebreaker to rule-in HIV infection. Instead, those with discrepant test results should be ruled inconclusive and referred for further testing in 14 days to rule-in or rule-out seroconversion.*
- *WHO recommends that all HIV testing algorithms use a combination of RDTs and/or EIAs to achieve at least 99% positive predictive value and use a combination of tests with $\geq 99\%$ sensitivity and $\geq 98\%$ specificity. WHO does not recommend the use of NAT techniques within HIV testing algorithms for individuals ≥ 18 months of age.*
- *The first test in an HIV testing strategy and algorithm should have the highest sensitivity, followed by a second and third test of the highest specificity. Algorithms should be validated and verified to ensure high quality and accurate testing services.*
- *All people newly diagnosed with HIV should be retested to verify their HIV status prior to starting ART, using the same testing strategy and algorithm as the initial test. To minimize the risk of misdiagnosis, this approach should be maintained in settings in which rapid ART initiation is being implemented.*

Source: WHO 2019, [6]

See revision on lines 145-150:

Data were extracted from each policy document by one reviewer (EF) into standardised coding forms on policy information, HIV testing strategy/algorithm (>18 months of age), dual HIV/syphilis testing strategy/algorithm, and retesting prior to ART initiation (online supplementary table 1). When reported the order, type of test kits and name brand of diagnostics used within the HIV testing algorithm was summarised descriptively, as well as information on whether the algorithm was verified or validated.

See revision on lines 158-166:

We assessed national adoption of WHO HTS guidance using six specific recommendations related to testing strategies and algorithms for those ≥18 months of age set forth in the WHO 2019 HIV testing guidelines, namely: (1) use of serial testing, (2) use of a 3-test strategy, (3) discontinuation of a tie-breaker to rule in HIV infection, (4) discontinuation of WB/LIA, (5) retesting prior to ART initiation, and (6) use of dual HIV/syphilis RDT in ANC. Additional qualitative details about the national testing algorithm, such as the order of and name brand of test kits used in testing algorithms, were also reviewed when reported and assessed according to WHO recommendations (see Box 1).

See revision on lines 409-425:

While countries will need to develop a plan and identify the optimal time for the transition, in terms of costs, a modelling study found that the total cost of the 3-test strategy would have a very limited impact on costs.[20] Further, in Ghana after transitioning to a 3-test strategy they found that while there were lessons learned about logistics and training, the new strategy was found to be feasible, improved testing quality and reduced costs due to misdiagnosis.[21]

Reviewer: 2 Dr. Nancy Puttkammer, University of Washington School of Public Health Comments to the Author: This paper presents results of a review of HIV testing policies in 45 African countries, following the WHO's 2019 HTS guidelines. It is a very useful contribution to understand how policy adoption has proceeded with respect to these guidelines.

Overall comments

- Major comment: I am concerned with the presentation of the results on longitudinal comparison of guideline compliance. It is problematic to make the claim of 2.4 fold improvement in policy compliance given that only 32 policies were reviewed in 2018 compared to 45 in 2021. Those not reviewed (unobserved) in 2018 could represent significant bias in the results. For example, if all 13 of the unobserved cases were actually compliant with adoption in 2018, this would have been a compliance level of $n=22/32=69\%$ -- similar to the level observed in 2021. It would be a fairer comparison longitudinally to compare the same 32 countries.

We have now adjusted the comparison between the 32 policies which were both included in the 2018 and 2021 reviews. This should address some of the bias. We did continue to report all the findings from the 2021 review but adjusted the comparisons.

See revisions lines 206-211:

When the analysis was restricted to a subset of countries with policies in both the 2018 and 2021 review ($n=32$), the proportion adopting WHO guidance in 2018 was 28% ($n=9/32$) vs 62% ($n=20/32$) in the current (2021) review (see online supplementary table 3; supplementary figure 1). Across the sub-regions, Southern (80%) and Eastern (75%) Africa had the greatest level of alignment, whilst Central Africa (63%) and Western Africa (44%) had the lowest adoption rates (table 1; figure 2).

Supplementary Table 3. Progress on policy adoption in the WHO African Region from 2018 to 2021

Subregion	Count	Country	Policy reviewed in 2018?	Adherence in 2018	Policy reviewed in 2021	Adherence in 2021
Central Africa	1	Angola	Yes, 2015	Not adopted	2020	Mostly adopted
	2	Cameroon	Yes, 2015	Not adopted	2019	Mostly adopted
	3	CAR	Yes, 2010	Not adopted	2018	Not adopted
	4	Chad	Yes, 2011	Not adopted	2017	Mostly adopted
	5	DRC	Yes, 2017	Adopted	2020	Adopted
	6	Equatorial Guinea	No	–	2018	Not adopted
	7	Gabon	No	–	2017	Mostly adopted
	8	Republic of the Congo	No	–	–	–
	9	São Tomé e Príncipe	No	–	2018	Not adopted
Eastern Africa	10	Burundi	Yes, 2016	Not adopted	2020	Mostly adopted
	11	Comoros	Yes, 2007	No information	2016	Not adopted
	12	Eritrea	No	–	2019	Not adopted
	13	Ethiopia	Yes, 2017	No information	2018	Mostly adopted
	14	Kenya	Yes, 2017	Mostly adopted	2021	Adopted
	15	Madagascar	Yes, 2011	No information	2018	Mostly adopted
	16	Malawi	Yes, 2016	Not adopted	2016	Mostly adopted
	17	Mauritius	No	–	2020	Not adopted
	18	Mozambique	Yes, 2016	Not adopted	2020	Mostly adopted
	19	Rwanda	Yes, 2016	Not adopted	2018	Mostly adopted
	20	Seychelles	No	–	2019	Not adopted
	21	South Sudan	Yes, 2017	Not adopted	2020	Adopted
	22	Tanzania	Yes, 2017	Not adopted	2021	Mostly adopted
	23	Uganda	Yes, 2016	Mostly adopted	2020	Mostly adopted
	24	Zambia	Yes, 2018	Not adopted	2020	Not adopted
25	Zimbabwe	Yes, 2016	Adopted	2016	Adopted	
Southern Africa	26	Botswana	Yes, 2016	Mostly adopted	2016	Not adopted
	27	eSwatini	Yes, 2018	Not adopted	2018	Not adopted
	28	Lesotho	Yes, 2016	Adopted	2021	Mostly adopted
	29	Namibia	Yes, 2016	No information	2018	Not adopted
	30	South Africa	Yes, 2016	Mostly adopted	2020	Not adopted
Western Africa	31	Algeria	Yes, 2013	Adopted	2013	Not adopted
	32	Benin	No	–	2017	Not adopted
	33	Burkina Faso	Yes, 2008	Not adopted	2021	Mostly adopted
	34	Cabo Verde	No	–	–	–
	35	Ghana	Yes, 2014	Not adopted	2019	Mostly adopted
	36	Guinea	No	–	2019	Mostly adopted
	37	Guinea-Bissau	No	–	2021	Not adopted
	38	Ivory Coast	Yes, 2016	Not adopted	2016	Not adopted
	39	Liberia	Yes, 2015	Not adopted	2020	Mostly adopted
	40	Mauritania	No	–	2020	Not adopted
	41	Mali	No	–	2017	Not adopted
	42	Niger	No	–	2020	Not adopted
	43	Nigeria	Yes, 2016	Not adopted	2020	Mostly adopted
	44	Senegal	Yes, 2017	Mostly adopted	2018	Not adopted
	45	Sierra Leone	Yes, 2017	Not adopted	2020	Not adopted
	46	The Gambia	Yes, 2014	Not adopted	2019	Not adopted
	47	Togo	No	–	2019	Not adherent

A total of 32 countries had policies reviewed in 2018 and 2021. Those highlighted in red are policies reviewed in 2018 that were updated (n=28). Number of policies deemed adopted in 2018 (n=7) and number deemed adopted in 2021 (n=20)

Supplementary figure 2. Comparing policy adoption across countries, 2018 vs 2021

- For readers who are less familiar with the criteria for moving from the 2-test to the 3-test strategy, the paper might benefit from presenting this information a bit differently. Lines 114-116 state: “However, as ART coverage expands and HIV positivity declines, WHO now recommends countries to use the national HTS positivity and the treatment-adjusted prevalence, which accounts for both the national HIV prevalence and ART coverage, as an indicator to determine when to transition to a three-assay testing strategy”. This sentence suggests that some countries should retain the two-assay testing strategy. However, the review paper seems to consider use of the two-assay strategy as non-compliant with the 2019 guideline. If there is some nuance on whether or not all countries should transition, perhaps this should not be considered as a criteria for adoption vs. non-adoption of the 2019 guidelines. However, the authors could still provide descriptive results on where guidelines stand as of 2021 with respect to this issue.

Thank you so much for this comment. We have added clarification here that all countries are advised to utilise a 3-test strategy for a positive diagnosis as HIV positivity declines due to more people receiving ART and fewer absolute numbers of undiagnosed people with HIV. Because HIV positivity among the population being tested directly affects the positive predictive value of testing algorithms, this recommendation applies to all settings.

See revision lines 99-109:

Additionally, many countries still use the 2015 WHO recommendation of using the national HIV prevalence to determine whether a two-assay ($\geq 5\%$) or three-assay testing strategy ($< 5\%$) should be used.[1] However, as ART coverage has expanded HIV positivity and the proportion of people with HIV who are undiagnosed and not in care will continue to decline. As a result, the positive predictive value of previous testing strategies will also decline and lead to an increase in false positive diagnoses. In relation to the guideline development process, WHO now recommends all countries use a standard 3-test strategy.[8] WHO also recommends that countries planning to update their HIV testing algorithms undertake a verification study to select appropriate HIV serology products and ensure they don't cross-react in order to minimize the risk of misdiagnosis.[9]

See revision within Box 1 which starts on line 110 (above) and line below:

- *In response to changes in the HIV epidemic, WHO recommends countries use three consecutive reactive tests to provide an HIV-positive diagnosis.*
- The authors are inconsistent in labeling and presenting the key guideline policy recommendations that are included in the 2019 HTS guidance. I recommend aligning the

phrasing used and the order in which the key policy recommendations are presented and discussed across the paper. For example, the areas of guideline compliance as listed in the abstract do not fully align with the results in Table 1 or in the subsections shown in the results section, in terms of order in which the items are shown. Lines 314-323 adds a sub-section on NAT, but this does not appear in Table 1 and the manuscript is not clear on the 2019 WHO guidelines with respect to NAT. Can this be folded into other sub-sections so that the results section stays aligned with the 6 areas of guidelines?

Thank you for this comment, we have added in a table summarising key recommendations related to this review in the 2019 guidelines. We have sought to clarify that where reported and possible we extracted information on the order and brand name of test kits used in algorithms. This now clarified in the methods and in Box 1 we've provided details. We've also clarified that this review does not specifically focus on NAT in testing algorithms and was limited to those. We have addressed some of this as well in the comments above.

See revision lines 145-150:

Data were extracted from each policy document by one reviewer (EF) into standardised coding forms on policy information, HIV testing strategy/algorithm (>18 months of age), dual HIV/syphilis testing strategy/algorithm, and retesting prior to ART initiation (online supplementary table 1). When reported the order, type of test kits and name brand of diagnostics used within the HIV testing algorithm was summarised descriptively, as well as information on whether the algorithm was verified or validated.

See also revision starting in Box 1 on line 110:

WHO does not recommend the use of NAT techniques within HIV testing algorithms for individuals ≥18 months of age where NAT-based algorithms

- I would like to see the authors situate their inquiry in the broader base of knowledge about effective policy dissemination and implementation. There is a body of literature on policy implementation science, guideline dissemination and implementation, and policy change process that would be very interesting to reference and discuss as part of the paper. I recommend that the background/introduction section include 2-3 sentences about how the 2019 WHO HTS guidelines were disseminated to countries and globally. Then, in the Discussion section, it would be helpful to situate the findings in terms the policy change processes in countries. Policy change is not easy, and requires multiple steps and stakeholder engagement in each country setting. The authors spend most of the discussion section focused on very specific issues about HTS without zooming out to discuss their recommendation that countries should update their guidelines and algorithms more rapidly or frequently. It would be helpful if they can put this recommendation in the context of acknowledging what it takes to make policy changes. The discussion of algorithm validation would fit well in this part of the discussion, recognizing that many countries feel a need to validate WHO's global recommendations in their own contexts, or at least go through a formal policy review and adoption process that considers if/how to adapt WHO guidelines to their settings. The following papers might provide useful grounding in policy implementation concepts and principles:
 - Nilsen P, Ståhl C, Roback K, Cairney P. Never the twain shall meet?--a comparison of implementation science and policy implementation research. *Implement Sci.* 2013 Jun 10;8:63. doi: 10.1186/1748-5908-8-63. PMID: 23758952; PMCID: PMC3686664.
 - Oh A, Abazeed A, Chambers DA. Policy Implementation Science to Advance Population Health: The Potential for Learning Health Policy Systems. *Front Public Health.* 2021 Jun 17;9:681602. doi: 10.3389/fpubh.2021.681602. PMID: 34222180; PMCID: PMC8247928.
 - Flodgren G, Hall AM, Goulding L, Eccles MP, Grimshaw JM, Leng GC, Shepperd S. Tools developed and disseminated by guideline producers to promote the uptake of their

guidelines. Cochrane Database Syst Rev. 2016 Aug 22;(8):CD010669. doi: 10.1002/14651858.CD010669.pub2. PMID: 27546228.

Thank you for this comment. We have incorporated these comments in the introduction.

See revision in lines 112-119:

To support the implementation of the 2019 guidelines, WHO launched and disseminated the guidance at the Africa Society of Laboratory Medicine and the International Conference on AIDS and STIs in Africa. Following the initial release, WHO also provided detailed country support to adopt the guidelines along with developing an application to access the guidelines more easily. In addition to dissemination and country support, [10] it is critical to monitor and track the implementation of WHO policy uptake to understand policy priorities and challenges overtime. Such tracking not only provides valuable insights but can help guide revisions to future guidance and can help target country policy support [11—12].

We have also added in a reflection on the process and engagement needed to enact policy change.

See revision in lines 116-119:

It's critical to monitor and track the implementation of WHO policy uptake as is an important way to understand policy priorities overtime. It also provides valuable insight to understand if there may be challenges with adopting specific recommendations so that adjustments to guidance can be made and so country support can be provided.

See revisions in lines 352-356:

This is particularly critical now as WHO is now updating HTS guidance in 2023 and it will be essential for WHO to continue policy tracking, [10] to allocate sufficient resources, to conduct implementation science research to understand bottlenecks hinder policy change and to support country-led technical working groups to drive policy change [11—12].

- The writing is a bit awkward in spots. Common typos include incorrect verb tense and other. I recommend further copy editing for concise and clear writing. For example, o Line 264-66 is presently written as: “Among high HIV-burden countries, only 18% of policies (n=2/11) recommended the use of a 3-assay testing strategy and this corresponded to two countries, Malawi and Zimbabwe, respectively.” It would be clearer and simpler to write “Only two out of eleven high HIV-burden countries, Malawi and Zimbabwe, recommended the use of a 3-assay testing strategy (18%).”

Thank you for this comment. I have revised the results and included the proposed sentence in revised lines 221-222: *Overall, only two out of eleven high HIV-burden countries, Malawi and Zimbabwe, recommended the use of a 3-assay testing strategy (18%).*

We have also addressed copy editing issues as suggested throughout manuscript.

- Line 277-279: “Sixty-five percent of the policies (n=15/23) recommending retesting prior to ART initiation, conduct retesting using the same national HIV testing algorithm in a new specimen run by a different operator in a different testing site” could be rewritten as ““Sixty-five percent of the retesting policies (n=15/23) recommended retesting using the same

national HIV testing algorithm in a new specimen run by a different operator in a different testing site”.

Thank you – we have revised the sentence. See line2 289-294:

Sixty-five percent of the retesting policies (n=15/23) recommended retesting using the same national HIV testing algorithm in a new specimen run by a different operator in a different testing site; 26% (n=6/23) did not specify how retesting was carried out and 9% of policies (n=2/23) utilized retesting strategies that differed from the national HIV testing algorithm, which included parallel testing of A1/A2 in Botswana and Malawi.

Specific comments Abstract:

- The n's used in the Results paragraph are a bit confusing. Sentence on lines 43-45 is not clear. “Using a 2-assay testing strategy was the most common form of non-adoption: 31% (n=14) and 22% (n=10).” It is not immediately clear if the n's shown there represent numerators or denominators.
- In the Abstract, I recommend focusing more on the descriptive results in 2021 rather than the longitudinal change in guidelines adoption from 2018 to 2021 (see reasons above). However, if the authors wish to present a longitudinal comparison for the same countries assessed at both time points, as part of the Abstract, this would be OK at the end of the results section of the Abstract.

Thank you for all of these comments we've revised the abstract with input from the BMJ Open editor to ensure it aligns with their guidance and format. We've also incorporated these comments and have focused on the status of policies based on the 2021 review.

See revision in lines 25-50:

- **ABSTRACT**

Objectives: In 2019, the World Health Organization (WHO) released guidelines on HIV testing services (HTS). We aim to understand adoption of these recommendations.

Design: Policy review

Setting: 47 countries within the WHO African region.

Participants: National HTS policies from the WHO African region as of December 2021.

Primary and secondary outcome measures: Uptake of WHO recommendations across national HTS policies including the standard 3-test strategy; discontinuation of a tie-breaker test to rule in HIV infection; discontinuation of western blotting (WB) for HIV diagnosis; use of retesting prior to ART initiation; and the use of dual HIV/syphilis rapid diagnostic tests (RDTs) in antenatal care (ANC). Country policy adoption was assessed on a continuum, based on varying levels of complete adoption.

Results: National policies were reviewed for 96% (n=45/47) of countries in the WHO African region, 40% (n=18) were published before 2019, and 62% (n=28) adopted WHO guidance. As of 2021, adoption in the region was 62% (n=28/45). Among countries that had not fully adopted WHO guidance, using a 2-test strategy was the most common reason for misalignment; 31% (n=14) and 22% (n=10) in low (<5%) and high (≥5%) prevalence countries, had not yet adopted the 3-test strategy. Ten policies (22%) recommended the use of WB in their HIV testing algorithm, and 51% (n=23) recommended retesting before ART initiation. Dual HIV/syphilis RDTs were recommended in 47% (n=21/45) of policies.

Conclusions: Many countries in the African region have adopted WHO-recommended HIV testing strategies. While WB was only used in a few countries, concerted efforts are needed to phase out this technology in favour of RDTs. Countries should accelerate their transition to WHO

recommendations by streamlining efforts to adopt and implement a 3-test strategy and dual HIV/syphilis RDTs.

Introduction

- In the paragraph starting on line 97, it would help to underscore the importance of the 2019 guideline changes if the authors could summarize in 2-3 sentences the advantages of the guideline changes. Does the evidence base for the changes demonstrate improved (or at least non-inferior accuracy), less drop-out in the HIV cascade, lower cost, greater feasibility or simplicity of HTS services, etc?

Thank you for this, we have added in a new paragraph to summarise some of the key priorities and reason for the recommendations.

See revision in lines 94-109:

These recommendations have been prioritised because they are essential to achieving global 95-95-95 goals by making testing accurate, affordable and high impact. [3,6] By moving away from testing services with WB, programmes will no longer have long turnaround times which delay the ability to offer same day ART or PrEP initiation. [6] Adoption of innovative tools such as the dual HIV/syphilis RDTs will enable more people to be tested and treated for syphilis, which is essential for achieving triple elimination goals. [6] Additionally, many countries still use the 2015 WHO recommendation of using the national HIV prevalence to determine whether a two-assay ($\geq 5\%$) or three-assay testing strategy ($< 5\%$) should be used.[1] However, as ART coverage has expanded HIV positivity and the proportion of people with HIV who are undiagnosed and not in care will continue to decline. As a result, the positive predictive value of previous testing strategies will also decline and lead to an increase in false positive diagnoses. In relation to the guideline development process, WHO now recommends all countries use a standard 3-test strategy.[8] WHO also recommends that countries planning to update their HIV testing algorithms undertake a verification study to select appropriate HIV serology products and ensure they don't cross-react in order to minimize the risk of misdiagnosis.[9]

• Methods

- Line 148-49: If something was marked as 'unclear' within the documents gathered, was further information sought from key informants to clarify?

Yes. We have added in a sentence to make this clearer.

Lines 155-156:

Reviewers than worked to contact key informant to provide further detail and clarity wherever possible.

• Results

- Line 193: See comment above about longitudinal comparison. It is not useful to contain the absolute numbers or relative increase in policies adopting WHO HTS guidelines, because the denominators were different in 2018 vs. 2021.

We have addressed this above.

- Line 270: word “respectively” is not needed.

Thank you, see revised on line 305-306.

- Line 287: typo “Easter Africa”

We have addressed. See revision on line 311: *Eastern Africa*

- Line 271-272 references guidelines on dual HIV/syphilis testing, and it seems this would be better placed in the subsection on that topic (line 291)

We have moved this sentence down to the HIV/syphilis testing sub-section. See revision on lines 305-312:

Adoption of dual HIV/syphilis RDT in ANC settings

Forty-seven percent (n=21/45) of policies supported the use of dual HIV/syphilis RDTs among pregnant women in ANC (figure 3_6). The region with the highest policy uptake was Southern Africa (60%), followed by Eastern (56%), Western (44%) and Central Africa (25%) (table 1). And of policies supporting a 3-test strategy, only 53% (n=11/21) also recommended the use of dual HIV/syphilis RDTs among pregnant women. The majority of policies (n=20/21) recommended the use of dual HIV/syphilis RDT as A1 among pregnant women and only one policy (Lesotho) recommended its use as a triage test (A0) as an interim algorithm while waiting for the results of a national verification study.

- Line 306: can you name the three countries recommending dual HIV/syphilis testing among groups outside of ANC?

Yes, this detail has been added. See revision on line 322-328:

In addition to the use in ANC settings, three countries also recommended testing with dual HIV/syphilis RDTs among other populations such as male partners of pregnant women and key populations (Liberia, Madagascar and Uganda). Only 50% of policies (n=11/21) specified the brand name of the dual HIV/syphilis RDT in the testing algorithm, and among the policies mentioning the brand name, Bioline HIV/Syphilis Duo (Abbott Diagnostics Korea Inc, Republic of Korea) was the most employed RDT (100%). Only 57% of policies (n=12/21) mentioned whether syphilis treatment was provided immediately after a reactive syphilis test result.

- Discussion

- Line 330: I think it is misleading to start the sentence with “However”. Your point should be that “as of the 2021 review, 40% of policies were published before 2019 which underscores the need to update national testing policies at more regular intervals.”

See revised on line 346-348:

As of the 2021 review, 40% of policies were published before 2019 which underscores the need to update national testing policies at more regular intervals.

- Line 339: typo “WHO’s” should be “WHO”

See revised on line 214: *WHO 2019 HTS recommendations*

- Line 343: “has” should be “have” (countries is plural)

See revised on line 375: *low resource countries in the WHO African region have*

- Line 361: “given this changes” should be “given these changes”

See revised on line 402: *Given these changes in the epidemic*

- Line 399: would be helpful to define the acronym NCPI and explain what is meant by this. The following sentence is unclear on the authors' interpretation of possible bias. Are they saying that policy reviews are never to be trusted? Or that their review is accurate while surveys of policy are not?

Thank you. Further detail on what NCPI is and the difference in policy reviews and policy surveys is now explicitly noted. See lines 428-436:

However, the number of countries adopting dual HIV/syphilis RDTs among pregnant women may be in practice higher considering that preliminary National Commitments and Policy Instrument (NCPI) survey data reported that a total of 26 countries in the WHO African region adopted the use of dual HIV/syphilis RDTs in ANC.[24] The differences likely reflect differences in methodologies (policy reviews versus policy survey). Often annual global policy surveys, like NCPI, may reflect a mixture of policies and anecdotal practices, including unofficial policies not yet within official guidelines. Although the findings of this review likely provide a conservative estimate of policy adoption, they do highlight the need to ensure official national policies are updated accordingly.

- Lines 414-416 are unnecessary in the sentence. Overall the paragraph on conducting algorithm verification studies could be shortened, as it is not the central focus of this paper.

See revision on lines 448-460:

As countries update their national HIV testing guidelines to incorporate latest WHO guidance and adapt their national HIV testing strategies and algorithms to shift toward using three consecutive reactive serology tests to diagnose HIV, introduce dual HIV/syphilis RDTs among pregnant women and key populations, move away from western blotting, reinforcing retesting prior to ART initiation as well as design flexible algorithms to address kit shortages, WHO encourages national programmes to conduct a verification study to provide objective evidence, before national scale-up, that a specific combination of products work well together without sharing false-reactivity, thus reducing the risk of misdiagnosis.[6, 9] To assist countries achieve this, WHO has put together a practical toolkit to accelerate policy adoption.[27] To date, the following countries are working directly with WHO to update their testing algorithm: Armenia, Burkina Faso, Cameroon, Chad, Central African Republic, Cote D'Ivoire, Democratic Republic of Congo, eSwatini, Kenya, Lao, Lesotho, Mali, South Sudan and Zambia.

- I recommend more discussion of the strengths and limitations in the body of the paper. The authors include bullet points but do not include these in the body of the Discussion section of the paper.

We appreciate this point. The bullet points are requested per the BMJ Open style guide and were noted again by the editor as part of reviewer comments. We have followed the journals advice here on their placement. We have kept the bullet points and included a paragraph as well as you suggest.

See revision line 462-477:

This policy review has strengths as well as weaknesses. Firstly, this review provides a comprehensive review of existing national HTS policies in the African region and was able to identify current policies for 45 countries. While this represents nearly all countries in the region, we were unable to identify policies for two countries. Secondly, unlike other policy surveys this study reviewed official policies for each country that were collected through a robust global search and triangulated data with additional information provided by country-level key informants. Despite this, the review was unable to assess unpublished policies and those in development at the time of the analysis. During the review some countries indicated that they were in the process of updating their guidance, particularly around adoption of a 3-test strategy. We were unable to include this information in the review, and thus our reported level of adoption of WHO recommendations may be an underestimate. Thirdly, this review did not assess policy implementation at the national, subnational, or site level. Thus, it is possible that some practices in country differ than policies reviewed, and level of adoption reported may vary. Lastly, we focused on HIV testing strategies for individuals ≥ 18 months of age. As a result, we were unable to report on the adoption of HIV testing strategy recommendations for children, such as early infant diagnosis.

- Figures
 - Figures 2 and 3 are helpful, but could be difficult for readers with red-green color blindness to interpret. Please consider using a different color contrast.

Thank you we have updated the colours accordingly.

See revised figure 2:

See revised figure 3:

Reviewer: 1 Competing interests of Reviewer: None

Reviewer: 2 Competing interests of Reviewer: I have no competing interests.

VERSION 2 – REVIEW

REVIEWER	Domingues, Rosa INI/Fiocruz, LAPCLIN AIDS
REVIEW RETURNED	26-Sep-2023

GENERAL COMMENTS	Dear editor, the study is relevant and the revised version is much clearer. I have small suggestions Abstract: The abstract can be improved Aims: "In 2019, the World Health Organization (WHO) released guidelines on HIV testing services (HTS). We aim to understand adoption of these recommendations". Suggestion: To assess adoption by African countries of six 2019 WHO recommendations related to HIV testing strategies. Results (lines 38-39): "...and 62% (n=28) adopted WHO guidance. As of 2021, adoption in the region was 62% (n=28/45)". Same information. Please review. Results (lines 40-41): "...using a 2-test strategy was the most common reason for misalignment". Using a 3-test strategy and use of dual HIV-syphilis test had the same adoption rate (47%). Please review. Results (lines 41-42): "...31% (n=14) and 22% (n=10) in low (<5%) and high (≥5%) prevalence countries, had not yet adopted the 3-test strategy". These numbers are not easy to understand, since among countries with high prevalence, only 2 (out of 11) adopted the 3-test strategy. Therefore, 81,8% have not yet adopted this recommendation. Please review. I suggest that the authors present the results using the same criteria. For example, for the use of WB, the authors report those who did not adopt the recommendation (22%), but for "retesting before ART initiation" and "use of dual syphilis/HIV RDT", the
---

	authors report the proportion of countries that adopted the recommendation. It is easier for the reader if the presentation of results follows the same pattern. Abstract Conclusion: the authors' conclusion does not focus on recommendations that have had lower adoption. For example, discontinuation of WB was the recommendation with the second highest adoption, but was the first mentioned in the conclusion. However, retesting before ART initiation was adopted in only 51% of policies, but was not mentioned in the conclusion. Please review. Lines 175/176: "Descriptive analyses were then stratified by subregions (Western, Central, Eastern and Southern Africa; online supplementary figure 1) were also conducted to determine rates of adherence by subregion". Please review. Lines 216-222: Please review the numbers. If two countries recommended the use of a 3-assay testing, then 09 countries among 11, and not 10 among 11, continued to use a 2-test strategy. Line 306: "The region was the highest policy uptake was Southern...". Please review. Line 332: The "Use of nucleic acid testing (NAT) to diagnose HIV infection in individuals ≥ 18 months of age" should not be included as a topic on the results section because this is not one of the six WHO recommendations assessed in the paper. Maybe this results should be presented in the topic "Adoption of serial testing strategy" or "Adoption of WHO standard 3-assay testing strategy" as the use of NAT was described as part of the testing algorithm. Line 362: "...respectively (save for discontinuation of western blotting)...". The sentence is not clear, please review. Lines 393-396: "Specifically, in high HIV burden settings in southern Africa (5 countries) and eastern Africa (6 countries), WHO recommends countries still using a 2-test strategy to prioritize moving toward using a 3-test strategy". As two countries (Malawi and Zimbabwe) have already adopted the 3-test strategy, perhaps the sentence should be more focused on the 9 countries that still use the 2-test strategy.
--	---

REVIEWER	Puttkammer, Nancy University of Washington School of Public Health, I-TECH, Department of Global Health
REVIEW RETURNED	28-Sep-2023

GENERAL COMMENTS	Thank you for addressing all the reviewer comments and recommendations. There remain a few minor spots where further copy editing is needed (such as using "don't" rather than "do not"). I believe these can be resolved in the final proofing stage.
--

VERSION 2 – AUTHOR RESPONSE

Dr. Rosa Domingues, INI/Fiocruz

Comments to the Author:

Dear editor, the study is relevant and the revised version is much clearer. I have small suggestions

Abstract: The abstract can be improved

Aims: "In 2019, the World Health Organization (WHO) released guidelines on HIV testing services (HTS). We aim to understand adoption of these recommendations".

Suggestion: To assess adoption by African countries of six 2019 WHO recommendations related to HIV testing strategies.

Thank you we have adjusted this to reflect the suggestion. Please see lines 26-28:

Objectives: In 2019, the World Health Organization (WHO) released guidelines on HIV testing services (HTS). We aim to assess adoption of six of these recommendations on HIV testing strategies among African countries.

Results (lines 38-39): "...and 62% (n=28) adopted WHO guidance. As of 2021, adoption in the region was 62% (n=28/45)". Same information. Please review.

We have revised accordingly to remove duplication in abstract. We have also aligned everything to the same denominator of n=47 to help with interpretation throughout the abstract to address comments about confusion. Please see lines 38-39:

Results: National policies were reviewed for 96% (n=45/47) of countries in the WHO African region, 38% (n=18) were published before 2019, and 60% (n=28) adopted WHO guidance.

Results (lines 40-41): "...using a 2-test strategy was the most common reason for misalignment". Using a 3-test strategy and use of dual HIV-syphilis test had the same adoption rate (47%). Please review.

We reviewed this data and this is correct, they do not necessarily reflect the same countries with alignment but the adoption rate is the same. We have provided a footnote to further clarify the findings on the dual test below table 1 on lines 228-231. We also revised the abstract sentence to reflect the denominator of 47, the absolute total countries in the region, to aid interpretation and address your comment more fully. See revision in lines 45-46.

See below a summary of related revisions:

Lines 45-46:

Dual HIV/syphilis RDTs were recommended in 45% (n=21/47) of policies.

Lines 234-236:

***Dual HIV/syphilis RDT as first test in ANC. Note that at the time of the review, five countries were using test outside of WHO guidance at the time. Two countries had non-compliant algorithms (Lesotho and Zambia) and three countries (Liberia, Madagascar and Uganda) were using the test outside pregnant women and among key populations and partners of pregnant women. However, as of 2023, use of the dual test is recommended among key populations and those with high ongoing risk.

Results (lines 41-42): "...31% (n=14) and 22% (n=10) in low (<5%) and high (\geq 5%) prevalence countries, had not yet adopted the 3-test strategy". These numbers are not easy to understand, since among countries with high prevalence, only 2 (out of 11) adopted the 3-test strategy. Therefore, 81,8% have not yet adopted this recommendation. Please review. I suggest that the authors present the results using the same criteria. For example, for the use of WB, the authors report those who did not adopt the recommendation (22%), but for "retesting before ART initiation" and "use of dual syphilis/HIV RDT", the authors report the proportion of countries that adopted the recommendation. It is easier for the reader if the presentation of results follows the same pattern.

We appreciate this note and have provided a general uptake number in the results. However, because the prevalence is important to understand the potential differences in adoption across countries, we have continued to provide this additional detail on high and low prevalence countries.

See revision in abstract in lines 40-44:

Among countries that had not fully adopted WHO guidance, not yet adopting a 3-test strategy was the most common reason for misalignment (45%, 21/47); of which 31% and 22% were in low- (<5%) and high-prevalence (\geq 5%) countries respectively.

Abstract Conclusion: the authors' conclusion does not focus on recommendations that have had lower adoption. For example, discontinuation of WB was the recommendation with the second highest adoption, but was the first mentioned in the conclusion. However, retesting before ART initiation was adopted in only 51% of policies, but was not mentioned in the conclusion. Please review.

We have revised the abstract to incorporate this into the conclusion and mainly focus on the policies with lower adoption.

See revision on lines 47-52:

Conclusions: Many countries in the African region have adopted WHO-recommended HIV testing strategies, however efforts are still needed to fully adopt WHO guidance. Countries should accelerate their efforts to adopt and implement a 3-test strategy, retesting prior to ART initiation and the use of dual HIV/syphilis RDTs.

Lines 175/176: “Descriptive analyses were then stratified by subregions (Western, Central, Eastern and Southern Africa; online supplementary figure 1) were also conducted to determine rates of adherence by subregion”. Please review.

This remains accurate. We have kept this so that readers can see how the regions were divided and which countries were in which region for clarity. Further, we provide several figures showing overall level of adoption by subregion and through a map to help illustrate this. This continues to be a part of the analysis presented in the paper. We have amended the language to help with grammar and clarity on lines 179-181:

Descriptive analyses were then stratified by subregions (Western, Central, Eastern and Southern Africa; online supplementary figure 1) to determine rates of adherence.

Lines 216-222: Please review the numbers. If two countries recommended the use of a 3-assay testing, then 09 countries among 11, and not 10 among 11, continued to use a 2-test strategy.

Thank you for this we have checked the database and identified that at the time of the review Malawi had not yet provided an updated testing algorithm, this was provided after the review had been completed. Thus, we have updated the text here and throughout the manuscript where relevant. See the revision according to lines below.

Lines 195-197:

Out of the six country policies that had not been fully updated since 2018 (Algeria, Botswana, eSwatini, Cote d'Ivoire, and Zimbabwe), one country (Zimbabwe) provided a new HIV testing algorithms that was included in the review.

Lines 223-228:

The most common reasons for not fully adopting WHO guidance was: continuing to use a 2-test strategy to make an HIV-positive diagnosis (n=9/11), employing both serial and parallel testing strategies (n=4/11), not adopting dual HIV/syphilis RDTs in ANC (n=4/11) and not implementing retesting prior to ART initiation (n=3/11). Overall, only two one out of eleven high HIV-burden countries, Malawi and Zimbabwe, recommended the use of a 3-assay testing strategy (18%).

Line 287-289:

Among high HIV-prevalence countries, only one out of eleven countries (9%) officially recommended the use of a 3-test strategy, which was Zimbabwe.

Lines 292-295:

Overall, across countries which had policies reviewed in both 2018 and 2021, 9 policies that previously recommended a 2-assay testing strategy transitioned to a 3-test strategy as of 2021 (Burkina Faso, Burundi, Chad, Ethiopia, Ghana, Kenya, Lesotho, Madagascar, and Zimbabwe, respectively).

Line 431-432:

This review found that only 9% (n=1/11) of the high-burden countries in the WHO African region has a supportive policy on a 3-assay testing strategy.

Line 438-442:

Many countries indicated they were in the process of transitioning to a 3-test strategy but needed to finalize guidance, complete verification studies, conduct tendering process for products, and garner consensus and support among partners. For example, after the review was completed, Malawi updated their guidance and currently recommends a 3-test strategy.

Line 306: "The region with the highest policy uptake was Southern...". Please review.

Thank you for this, we have revised it accordingly. See revision on lines 330-331:

The region with the highest policy uptake was Southern Africa (60%), followed by Eastern (56%), Western (41%) and Central Africa (22%) (table 1).

Line 332: The "Use of nucleic acid testing (NAT) to diagnose HIV infection in individuals ≥ 18 months of age" should not be included as a topic in the results section because this is not one of the six WHO recommendations assessed in the paper. Maybe this results should be presented in the topic "Adoption of serial testing strategy" or "Adoption of WHO standard 3-assay testing strategy" as the use of NAT was described as part of the testing algorithm.

Noted we removed the section and have moved the text up to be within the section on the "Adoption of the WHO standard 3-assay testing strategy". See lines 297-305:

Further, we identified that twenty-one percent of national policies (n=10) recommended the use of NAT as part of the testing algorithm for the diagnosis of HIV infection in individuals ≥ 18 months of age. Four policies were from Eastern Africa (Malawi, Mauritius, Mozambique and United Republic of Tanzania); three national policies from Southern Africa (Eswatini, Lesotho and Namibia); two policies from Central Africa (Angola and Sao Tomé et Príncipe); and one national policy from Western Africa (Guinea Bissau). While generally not in alignment with WHO guidance, two policies employed NAT as the third assay (A3) of the algorithm while the remaining eight policies recommended NAT to resolve persistent inconclusive/indeterminate test results (referring either to viral load, PCR or DNA-PCR).

Line 362: "...respectively (save for discontinuation of western blotting)...". The sentence is not clear, please review.

We have revised accordingly to make the sentences clearer. See revision on lines 385-389:

One likely explanation for the higher policy adoption is that these WHO's recommendations were first published between 2012 and 2015, with the exception of the discontinuation of western blotting which was recommended in 2019. Thus, national HIV programmes have had more time to fully incorporate these recommendations into their national guidelines.

Lines 393-396: "Specifically, in high HIV burden settings in southern Africa (5 countries) and eastern Africa (6 countries), WHO recommends countries still using a 2-test strategy to prioritize moving toward using a 3-test strategy". As two countries (Malawi and Zimbabwe) have already adopted the 3-test strategy, perhaps the sentence should be more focused on the 9 countries that still use the 2-test strategy.

Thank you for this. We have addressed this by focusing on countries that have yet to adopt the recommendation as you suggested by adding an additional sentence. See revision on lines 427-429:

Transition to a 3-test strategy should be a priority, particularly for the high HIV burden countries that were using a 2-test strategy at the time of this review.

Reviewer: 2

Dr. Nancy Puttkammer, University of Washington School of Public Health

Comments to the Author:

Thank you for addressing all the reviewer comments and recommendations. There remain a few minor spots where further copy editing is needed (such as using "don't" rather than "do not"). I believe these can be resolved in the final proofing stage.

Thank you so much. We have reviewed for grammar and editing and made changes as suggested throughout to enhance the editing. For example, as you suggested we have changed areas where the word "don't" is used to "do not". See revision on lines 111-112:

serology products and ensure they do not cross-react in order to minimize the risk of misdiagnosis.[9]